# Flexible iontronics based on 2D nanofluidic material

Di Wei [1] ✉, Feiyao Yang [1], Zhuoheng Jiang[1] & Zhonglin Wang[1,2] ✉

Iontronics focuses on the interactions between electrons and ions, playing essential roles in most processes across physics, chemistry and life science. Osmotic power source as an example of iontronics, could transform ion gradient into electrical energy, however, it generates low power, sensitive to humidity and can't operate under freezing point. Herein, based on 2D nanofluidic graphene oxide material, we demonstrate an ultrathin (~10 μm) osmotic power source with voltage of 1.5 V, volumetric specific energy density of 6 mWh cm$^{-3}$ and power density of 28 mW cm$^{-3}$, achieving the highest values so far. Coupled with triboelectric nanogenerator, it could form a self-charged conformable triboiontronic device. Furthermore, the 3D aerogel scales up areal power density up to 1.3 mW cm$^{-2}$ purely from ion gradient based on nanoconfined enhancement from graphene oxide that can operate under −40 °C and overcome humidity limitations, enabling to power the future implantable electronics in human-machine interface.

In biological cellular membranes, ion-specific pores permit certain types of ions flow across the membrane driven by osmotic energy from ion gradient, responsible for nervous impulses, muscle contractions and physiological sensing. The osmotic energy could be generated based on either pressure-retarded osmosis (PRO) or reverse electrodialysis (RED)[1], and the ion regulation component is the critical part for such power generation[2]. Iontronics couple the electron/ion charge transfers and exchange signals at the interface of electronic/ionic conductors, differentiating them from most electronics using just electrons and/or holes as the dominating charge carriers[3–7]. Bioinspired nanofluidic iontronics could have compatible signals with neurons to enable implantable iontronic devices or even neuronal-computer interfaces[3]. Enhanced sensitivity of tactile sensor[8] and pressure sensor[9] could also be obtained by iontronic films. Unusual behavior of ion transport kinetics in channels narrower than the Debye length of electrolyte has been observed, the surface charges on the inner walls of nanofluidic channels repel ions of the same charge and attract counter ions, making them the dominating charge carriers[10]. Such unipolar ion transport can enhance ionic conductivity up to several orders of magnitude, which breaks the conventional continuum-based theory[11]. Recently, a variety of unusual ionic phenomena such as highly selective ion sieving[12], ultrafast ion transport[13],

and anomalous increase of capacitance in nanopores[14,15] were all observed due to the enhanced diffusion related to the strong ion-ion correlations under severe nanoconfinement[16].

Graphene oxide (GO) as 2D nanofluidic material with negative surface charge from carboxyl and hydroxyl groups etc. has shown special affinity to water[17] and controllable ion transport properties[18]. It had also been reported to provide nanoconfined charging dynamic medium to be potentially used in many applications such as molecular sieving with ultrafast speed[13] and voltage gating devices[19] etc. The unipolar ion transport within 2D nanofluidic material and asymmetric charge distribution could be used to generate osmotic energy[20,21]. The asymmetric charge distribution could be introduced by wettability gradient[22] as observed in the power generation from GO and reduced GO junctions under moisture[23], as well as by induction of oppositely charged bilayer polyelectrolyte film that could generate a peak of 1.38 V at relatively humidity (RH) of 85%[24]. Osmotic energy originated from such charge gradient and regulated ion transport[25] in nanoconfined structures has been observed by examples from nanostructured carbon materials[26], graphene single microelectrode[27], $MoS_2$ nanopores[28], boron nitride nanotube[25], nanostructured silicon[29], protein nanowires[30] to membranes based on $MoS_2$[31], cellulose[32], silk[33] and MXene/Kevlar nanofiber composites[34] etc. Among above examples,

---

[1]Beijing Institute of Nanoenergy and Nanosystems, Chinese Academy of Sciences, 101400 Beijing, People's Republic of China. [2]School of Materials Science and Engineering, Georgia Institute of Technology, Atlanta, GA 30332, USA. ✉e-mail: weidi@binn.cas.cn; zlwang@binn.cas.cn

nanofluidic channels with tailored ion transport dynamics could enable high-performance RED. However, their power densities are generally small, ranging from $10\,\mu W\,cm^{-3}$ to $4\,mW\,cm^{-3}$ in recent reports[30,35]. In addition, fabrication of osmotic power source based on nanofluidic materials typically relies on expensive deposition/lithography techniques or nanoporous templates with sophisticated growth and processing steps. Such fabrication methods usually have better defined geometries, but limited applications due to the cost and sophistication. Jiang et al.[36] reviewed progress on enhanced ion regulation in nanofluidic devices for osmotic energy conversion. The essential challenges on their real world applications lie in the small energy and power densities, operation limitations on humidity and temperature, and feasibility of large-scale production.

Here, we show a flexible, ultrathin and printable GO-based triboiontronics and osmotic energy power source based on the ion gradient and the fine-tuned interfacial electrochemical reactions. To maximize the ionic power, a modular design osmotic energy power source with current enhancement purely from ion gradient is made from GO aerogels and self-healing ionogels based on room temperature ionic liquids (RTILs). The GO-based osmotic energy device and triboiontronics developed in this paper could operate under harsh environment regardless of low humidity and subzero temperature.

## Results

### Planar ultrathin osmotic power source based on 2D nanofluidic material of GO

Planar confinement in 2D nanofluidic material of GO expands translational degrees of freedom for ionic transport engendering unusual ion dynamics and ions transport much faster in the horizontal direction within 2D nanofluidic channels than in the vertical direction in the GO film[14,15]. Graphene nanopores[37] were found to preferentially transporting $K^+$ over its counter anions such as $Cl^-$ with selectivity ratios over 100 and hydrated $K^+$ diffuses orders magnitude more quickly than most hydrated ions[37] within the 2D nanofluidic channels. It was reported that potassium hydroxide (KOH) could partially remove the oxygen-containing groups of GO sheets through a series deoxygenation reaction leaving cations between graphene layers[38,39]. FTIR (Fourier-Transform Infrared Spectrometer) characterization and the related empirical structural formula of GO and rGO were shown in Supplementary Fig. 1. A reduction in the amount of hydroxyl and carboxyl groups happens to the structure of GO after addition of KOH, forming the reduced GO (rGO). $K^+$ was observed to migrate from rGO to GO when solid-state GO/rGO junction was formed at ambient humidity environment in our previous work[40]. The electrical conductivity of rGO was measured to be about $0.06\,S\,m^{-1}$, falling in the conductivity range of rGO by KOH treatment ($0.02–1.55\,S\,m^{-1}$)[38,39]. In this paper, rGO containing large amount of $K^+$ was chosen as the cation reservoir in the iontronic device. The schematic of the planar osmotic power source and its mechanism is shown in Fig. 1a, GO ink was deposited on one side of the electrode and rGO ink was deposited on the other, overlapping to form a junction. To avoid interference reactions, gold (Au) electrode[41] was used as charge collectors and $K^+$ will transport from rGO through 2D nanofluidic channels of GO to the cathode side of Au electrode under humidity. The scanning electron microscopy (SEM) image (Fig. 1b) clearly shows the cross section of stacked layers of GO formed from the ink is similar to that of the GO paper made from the filtration[16,42]. As shown in Supplementary Fig. 2a, the zeta potential of the GO solution reveals GO was negatively charged due to the abundant oxygen-containing functional groups. When forming the GO film from GO solution, only restacking of GO sheets happens and there are no chemical reactions involved and these oxygen-containing functional groups would not be removed. Thus, the surface of the GO flake should also be negatively charged. Ionic conductivity as a function of salt concentration was measured and the plateau in

Supplementary Fig. 2b indicated the $K^+$ could transport through the 2D nanofluidic channels inside the deposited GO film. Similar conductivity curve was also observed under the experimental condition with the $CaCl_2$ solution (Supplementary Fig. 2c). Such nonlinear conductance matched the previous reports from J. Huang et al. [11], and was verified to be originated from the cation transport through the 2D nanofluidic channels inside GO instead of from contaminations. SEM image in Fig. 1c shows that the cross section of rGO layer was different and some salt crystals seemed to be embedded inside the layered structure. The atom force microscopy (AFM) reveals different topographic morphology between GO (Supplementary Fig. 2d) and rGO (Supplementary Fig. 2e). The GO film had twisted flake-like topography, while the rGO film presented aggregates to form a more porous structure. The surface profile of the osmotic power source was shown in Supplementary Fig. 3. The dotted white line in the photograph shows where the probe took place. The total thickness of the coating was about $10\,\mu m$, including the thickness of the charge collector.

The current-voltage ($I–V$) characteristics of the osmotic power sources based on 2D nanofluidic material exhibited strikingly nonlinear effects, which may be due to the asymmetric transport of $K^+$ through the GO and rGO junction (GO/rGO). The open circuit voltages ($V_{OC}$) and short circuit currents ($I_{SC}$) could be obtained by reading the intercepts on the current and voltage axes by applying a sweeping voltage from 2 V to −2 V. $V_{OC}$ of the Au/GO/rGO/Au power source was about 1.2 V at room temperature under RH around 70% as shown in Supplementary Fig. 4a. Although the voltage of such osmotic power source was high, the current was low as it purely came from the ion gradient and there was no Faradaic reaction when using Au as charge collectors, however, addition of RTILs was found to boost up the current[8,9,40]. RTILs are molten salts with a melting point close to or below room temperature[43]. Unlike organic electrolytes, RTIL will not evaporate away in the encapsulated power source and it can also significantly accelerate charging dynamics in nanopores[44]. Different RTILs, 1-butyl-3-methylimidazolium bis(trifluoromethanesulfonyl) imide (BMIMTFSI), triethylsulfonium bis(trifluoromethylsulfonyl) imide (TESTFSI) and 1-ethyl-3-methylimidazolium tetrafluoroborate (EMIMBF$_4$) were tested, and they have increasing ionic conductivity at room temperature. TESTFSI was finally chosen for the development of anti-freezing osmotic power source since it has a relatively high ionic conductivity of $7.1\,mS\,cm^{-1}$ with a low melting point of $−35.5\,°C$[45]. Addition of RTIL enhanced the current but the $V_{OC}$ of Au/GO/RTIL/ rGO/Au kept the same around 1.2 V (Supplementary Fig. 4a), thus the current increase from RTIL may come from the enhanced ionic conductivity instead of redox reactions. Supplementary Fig. 4b also did not show any redox peaks coming from addition of RTIL and only capacitive currents existed in the cyclic voltammogramme (CV). The multimeter measurement verified consistent voltage of around 1.2 V with and without RTIL in osmotic power source (Supplementary Fig. 4c). Inspired from RED, electrochemical redox reaction was introduced at the interface between GO and Au charge collector to increase the current. Saturated silver nitrate (AgNO$_3$) aqueous solution was sprayed on top of Au at the cathode side before deposition of GO. The multimeter measurement showed voltage reached around 1.5 V (Supplementary Fig. 4d) unaffected by RTIL. The linear sweep voltammogramme in Supplementary Fig. 4e showed the $V_{OC}$ and $I_{SC}$ of Au/GO/RTIL/rGO/Au was 1.2 V and 1.4 $\mu A$, respectively. In comparison, the $V_{OC}$ and $I_{SC}$ of Au/AgNO$_3$/GO/RTIL/rGO/Au power source was 1.5 V and 11 $\mu A$, respectively. $I_{SC}$ increased almost 10 times with addition of AgNO$_3$, and the $V_{OC}$ of 1.5 V was consistent with that measured directly by the multimeter. The introduction of AgNO$_3$ significantly enhanced the energy and power density of the osmotic power source. Ionic rectification from the 2D nanofluidic materials is the origin of the osmotic energy. The nonlinear $I–V$ characteristic of the devices in Supplementary Fig. 4e in the negative voltage range, may be due to the

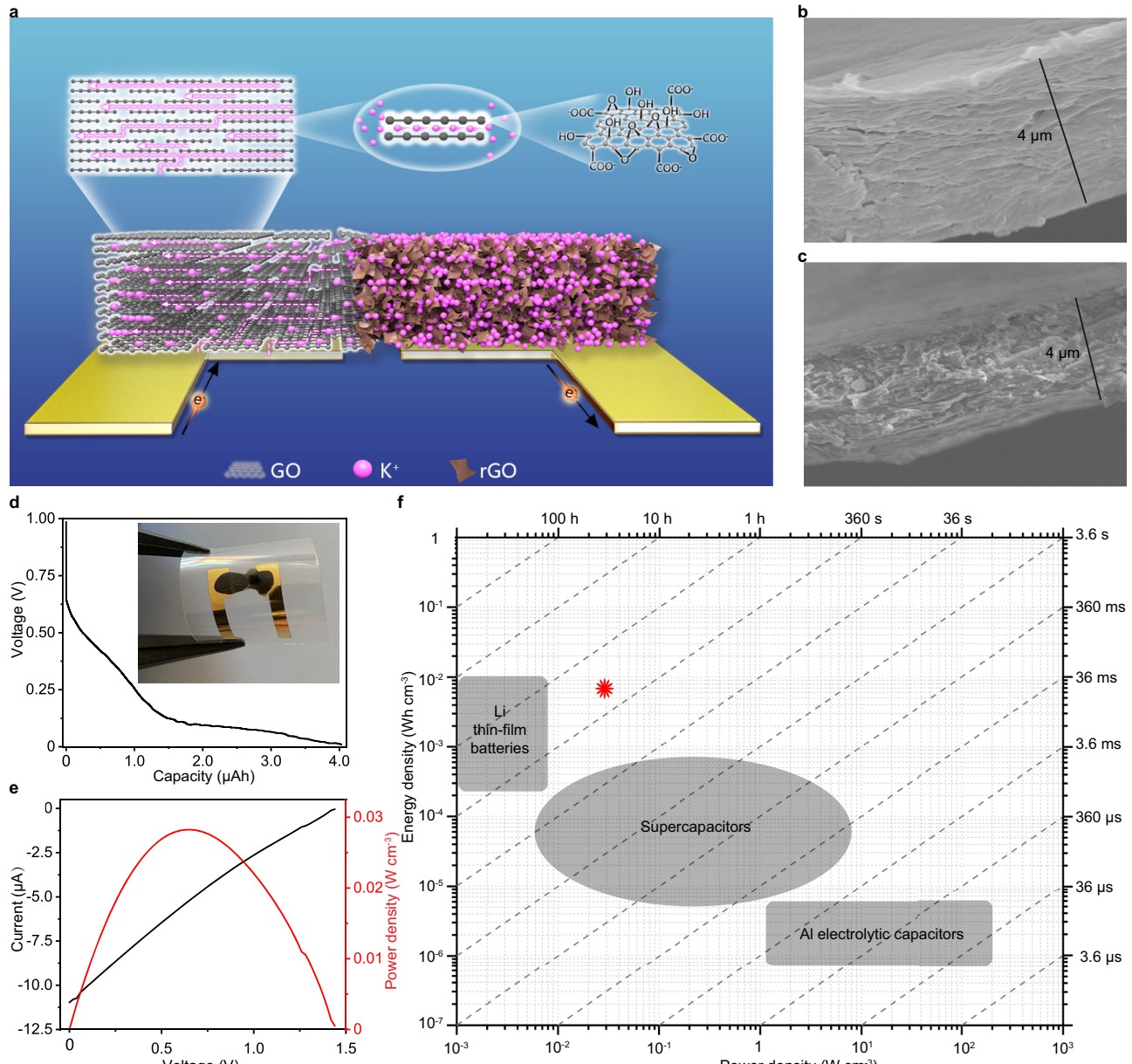

**Fig. 1 | The planar osmotic power source. a** Schematic illustration of GO/rGO junction and its mechanism of K⁺ conduction in 2D nanofluidic channels within the planar osmotic power source. **b** SEM image of the GO layer (cross section). **c** SEM image of the rGO layer (cross section). **d** Galvanostatic discharge performance of Au/AgNO₃/GO/RTIL/rGO/Au at 0.1 μA. Inset picture shows the photograph of the osmotic power source. **e** I−V characteristics and power density of Au/AgNO₃/GO/RTIL/rGO/Au. **f** Comparison of energy and power density of Au/AgNO₃/GO/RTIL/rGO/Au in the Ragone plot (Data for lithium thin-film batteries, supercapacitors and Al electrolyte capacitors are collected from refs. 47, 48). Source data are provided as a Source Data file.

ionic rectification, i.e., facilitation of K⁺ cation transport through negatively charged 2D nanofluidic channels of GO material under humidity. To investigate the possibility of the electrolytic water reactions, we carried out cyclic voltammetry test for the osmotic power sources with AgNO₃ (Au/AgNO₃/GO/RTIL/rGO/Au) and without AgNO₃ (Au/GO/RTIL/rGO/Au). It was shown that there is no Faradaic current coming from electrolysis of water in a sweep voltage range between −2 V to 2 V (Supplementary Fig. 4f) in both devices. Although the theoretical voltage of electrolysis of water is 1.23 V, the practical voltage for water electrolysis is generally much higher than that and is influenced by many parameters such as pH, salt concentration etc. As Supplementary Fig. 4g showed, the $V_{OC}$ of the Au/AgNO₃/GO/rGO/Au power source may consist of both the diffusion potential ($E_{diff}$) generated from ion gradient and the redox potential ($E_{redox}$) from electrochemical reactions. $E_{redox}$ may come from the reduction of Ag⁺ on

the cathode side between GO and Au interface, which is about 0.4 V as calculated in the Supplementary Discussion. $E_{diff}$ comes from harvesting the Gibbs free energy existing in the ion gradient and it can be estimated by the Nernst equation. Owing to the large salt intake and unimpeded water permeation, highly concentrated solutions that were close to saturation were formed within rGO, maintaining a large concentration gradient and enabling ultrafast ion permeation of hydrated K⁺ from rGO to GO driven by the gradient through 2D nanofluidic channels. The planar osmotic power source is in a solid-state form and it is impractical to calculation the concentration, however, the $V_{OC}$ of 1.2 V from Au/GO/rGO/Au power source came mainly from $E_{diff}$ since there were no redox reactions in it. Thus, the theoretical $V_{OC}$ of the Au/AgNO₃/GO/rGO/Au power source is calculated to be 1.6 V that matches the measurement of 1.5 V. It also provides a paradigm where the $V_{oc}$ of the osmotic power source could be tuned by tailoring interfacial

electrochemical redox reactions. Electrochemical impedance spectrum (EIS) is capable of high precision and is frequently used for the evaluation of heterogeneous charge-transfer parameters. The Nyquist plot (Supplementary Fig. 4h) of the osmotic power source with and without RTIL was compared and fitted by the most commonly used equivalent circuit for porous electrodes (inset of Supplementary Fig. 4h). Addition of RTIL improves not only the solution (contact) resistance but also the charge-transfer resistance significantly as shown in Supplementary Table 1. Furthermore, electrochemical impedance spectrum (EIS) confirmed the charge-transfer resistance decreased significantly with addition of RTIL as shown in Supplementary Fig. 4h. To reduce the influence from the diffusion controlled current caused by mass transfer, slow scan rate of $0.1 \, mV \, s^{-1}$ was used to reveal more information on the electron transfer process as the CV illustrated in Supplementary Fig. 4i. The device made of pure GO without ion gradient (Au/GO/RTIL/GO/Au) was used as benchmark, and it shows that whenever GO and rGO junction was formed (in case of Au/GO/RTIL/rGO/Au and Au/AgNO$_3$/GO/RTIL/rGO/Au), strong ionic rectification-like curve appeared below 0.2 V. The largest ion current from the Au/AgNO$_3$/GO/RTIL/rGO/Au may be due to the combination of ion gradient current from $E_{diff}$ and the Faradaic current from $E_{redox}$. Inset of Supplementary Fig. 4i demonstrated an obvious peak that may be correlated to the Ag$^+$/Ag redox potential. It seemed that 2D nanofludic channels enabled the confined electrochemical reactions at the interface between GO and Au charge collector. The image in Supplementary Fig. 5a, b also demonstrated the change in the AgNO$_3$ coating underneath GO on the cathode side, from crystal salts of AgNO$_3$ before discharge reduced to metallic silver particles after multi-cycles of discharge.

To evaluate the energy density of the ultrathin flexible Au/AgNO$_3$/ GO/RTIL/rGO/Au osmotic power source, it was discharge at $0.1 \, \mu A$. The discharge profile in Fig. 1d shows the voltage decades with time and energy capacity is calculated to be $0.69 \, \mu Wh$ as shown in Supplementary Fig. 5c. Although the capacity decreased within the first 5 charge–discharge cycles (Supplementary Fig. 5d), the results confirmed that the osmotic power source could be regenerated and its total weight is only 30 mg. Addition of AgNO$_3$ enabled the partially reversible redox reactions, similar to the process in RED systems[46]. The directional ion migration was then converted to electron transportation through redox reactions at the electrode surface. RTIL helps to accelerate K$^+$ transport in nanoconfined structures. The calculated maximum volumetric specific power density of $28 \, mW \, cm^{-3}$ from $I$–$V$ characteristics (Fig. 1e) is as high as supercapacitors and in good accordance with the measurement of output power as function of load resistance (Supplementary Fig. 5e). The maximum volumetric specific energy of the planar osmotic power source is $6 \, mWh \, cm^{-3}$ that is comparable to lithium thin-film batteries[47,48] as shown in the Ragone plot in Fig. 1f.

All batteries started discharge as soon as they were assembled, and this is particularly true for the concentration-gradient-based osmotic power source. However, it is also the unique benefit of the osmotic power source in that if it is packed separately or stored in vacuum or in a condition without any humidity, it will keep its energy and extend its shelf-life as long as we wish. Experiments has been carried out and our osmotic power source could not generate any power ($V_{oc}$ is zero) in the glovebox (Supplementary Fig. 5f) where the content of water could be ignored (H$_2$O < 0.5 ppm). It had been kept in the glovebox for half year, but when it was taken out, the $V_{oc}$ is still around 1.4 V (Supplementary Fig. 5g), which almost reached the $V_{oc}$ (1.5 V) as the freshly made osmotic power source (Supplementary Fig. 4d). Its power density kept value of $26 \, mW \, cm^{-3}$ (Supplementary Fig. 5h) comparable to the freshly made osmotic power source ($28 \, mW \, cm^{-3}$ in Fig. 1e). In addition, besides accelerating the kinetics of ion diffusion in nanoconfined channels, the RTIL itself used in the device is very stable at ambient environment with a broad electrochemical window. Unlike organic or aqueous electrolyte in batteries, RTIL will not evaporate away from the device.

## Conformable trationtronic device

The ultrathin and printable osmotic power source could be readily integrated with energy harvesting triboelectric nanogenerator (TENG) to form a self-charging trationtronic device. Iontronics are effective in modulating electrical properties through the electric double layer (EDL) assisted with ion migration. GO with abundant K$^+$ can realize a faster migration of cations under electric field to form the EDL. Through EDL capacitive coupling, a trationtronic energy harvesting and storage device was demonstrated. This device utilizes triboelectric potential and current originated from mechanical displacement to charge the osmotic power source. Owing to the universal existence of the triboelectricity, TENG is promising for applications in scavenging small mechanical energy from human activities. In our TENG unit, polyimide (PI) tape was chosen as the triboelectric material, which also served as the encapsulation layer. Rectangle copper foils with dimensions of $2 \times 2 \, cm$ by $50 \, \mu m$ thickness was attached on the PI tape as electrode. The total thickness of the TENG was <200 μm and it could be connected to the osmotic power source to form an ultrathin device that is conformal to human body (Fig. 2a). In order to elucidate the electrical output performances of the TENG unit, a linear motor was used to provide periodic contact-separation motion and a rabbit fur was used to be the contact material. With a rectifier in the integrated trationtronic device as shown in Fig. 2b, the negative voltage and current can be reversed into positive as shown in Fig. 2c, d. The generated voltage and current of the TENG were tested under 3 different frequencies, and the maximum voltage and current generated is 45 V and $0.7 \, \mu A$, respectively, at 3 Hz. The energy generated from such TENG was able to power a 1.5 V LED (Supplementary Movie 1). The voltage profile of the integrated energy harvesting (TENG) and storage trationtronic device was shown in Fig. 2e. When the osmotic power source was discharged to about 0.7 V, TENG can charge it back to 1.2 V in 200 s repeatedly. During sports, the motion frequency of human body may be even higher than 3 Hz, and larger energy generation could be expected. Such self-charging trationtronic device can be directly pasted on the surface of the item, and capable of adapting to any amorphous curved surface of supporting objects in any irregular shape with its conformability and bendability. Furthermore, the $V_{oc}$ and $I_{sc}$ under different humidity conditions, RH of 20%, 40%, 60%, 80% were tested for the thin TENG film pasted on the skin as shown in Supplementary Fig. 6, respectively. It was shown that the output performance of the TENG is quite stable under ambient temperature and humidity conditions. Only in the case of very high humidity conditions (RH higher than 80%), the output performance of the TENG will be slightly reduced. This proves that when the TENG is pasted on human skin, it can operate normally and the output performance is stable.

## Osmotic power source with nanoconfinement enhancement in 3D

To extend the nanoconfinement enhancement in 3D, we then further developed a nanoporous GO aerogel-based osmotic power source with self-healing ionogel electrolytes. Two GO aerogels with different ion concentrations were sandwiched between charge collectors and separated by an ionogel consisting of RTIL and polymer matrix. GO aerogel was fabricated through a self-assembled chemical reduction and freeze-drying process[49] as shown in Fig. 3a. SEM images in Fig. 3b showed the ordered interconnected network within the GO aerogel. For the GO + KOH aerogel, GO hydrogel was soaked in KOH aqueous solutions before freeze-drying to introduce ion concentration difference. The uniform distribution and high concentration K$^+$ were confirmed by the SEM images and energy dispersive X-Ray spectroscopy (EDS) results (Fig. 3c). The element composition analysis shows that the K$^+$ concentration in GO + KOH aerogels could reach up to 17.3%,

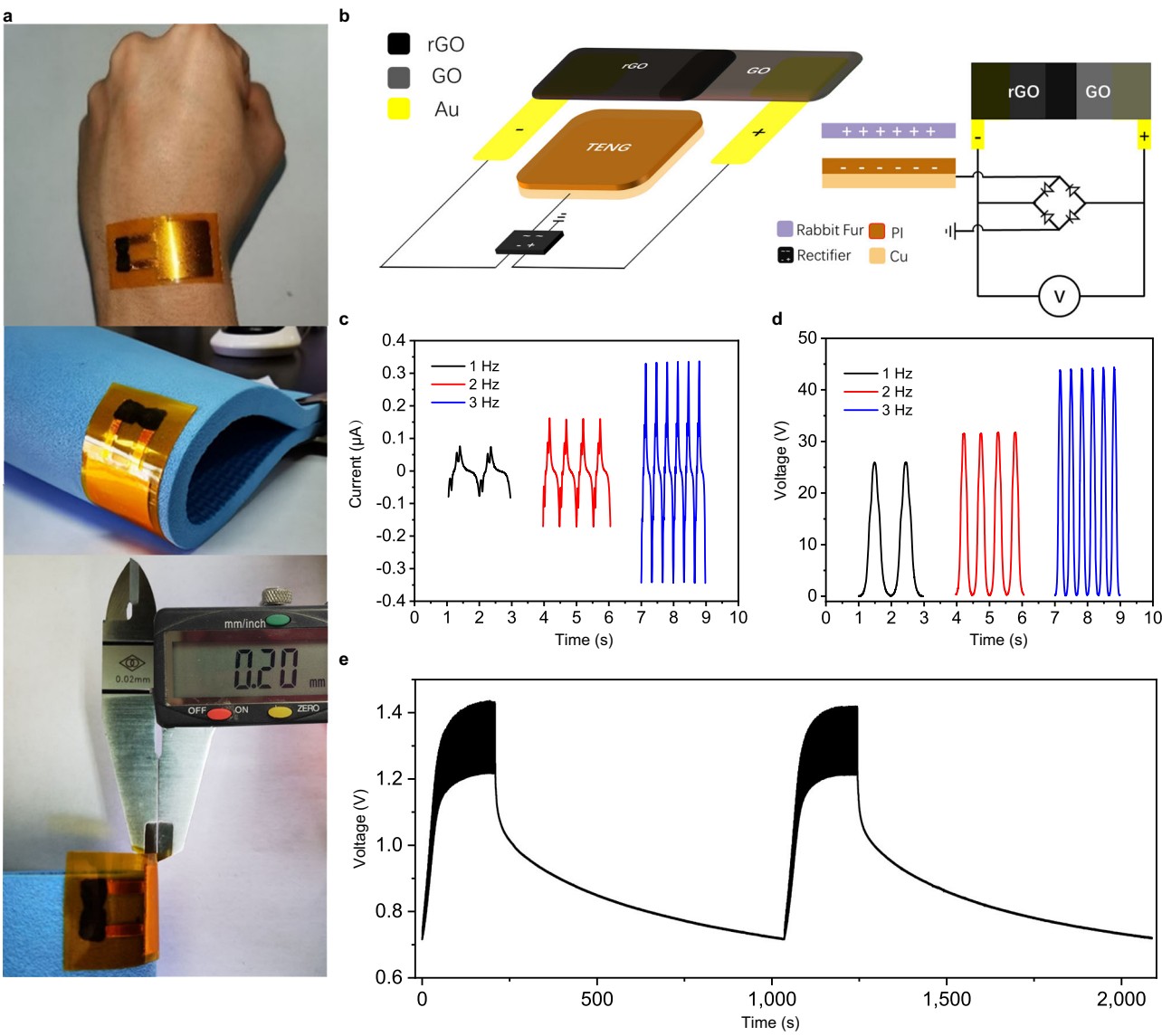

**Fig. 2 | Design and characterization of the triboiontronic device. a** The integrated energy harvesting (TENG) and storage (osmotic power source) triboiontronic device is bendable and its thickness is around 200 μm. **b** Structure of the integrated device. **c** Current and **d** Voltage generated by the TENG. **e** Voltage profile of the integrated self-charging triboiontronic device. The planar osmotic power source (Au/AgNO₃/GO/RTIL/rGO/Au) could be charged by TENG at frequency of 3 Hz and discharged at 1 μA. Source data are provided as a Source Data file.

owing to the 3D architectures of the aerogel and the large salt intake of GO materials. To enhance the physical stability and control the size of nanopore, the GO aerogel was made by partially reduction under a mild condition in the presence of leukocyte ascorbic acid (LAA). GO aerogels with different GO:LAA ratio was fabricated to investigate the influence of the LAA concentration. As shown in Fig. 3d, Raman spectroscopy was employed to study the disorder degree of GO materials. The peak located near 1350 cm⁻¹ is the disorder-induced band (D peak) and the peak close to 1580 cm⁻¹ is the graphite band (G peak)[39]. The area ratio of two peaks ($I_D/I_G$) decreases from 2.15 to 1.55 with increasing LAA concentration, meaning the structure disorder of GO aerogels decreases with increasing reduction degree. This is consistent with the morphology characterized by SEM (Supplementary Fig. 7a), which shows increasing sheet stacking and densely packed interconnected GO network with increasing LAA concentration. The surface area of the porous GO aerogel and the pore distribution were characterized by nitrogen adsorption/desorption measurements as shown in Fig. 3e, f. The Brunauer–Emmett–Teller (BET) surface area decreased from 403.5 m² g⁻¹ to 15.6 m² g⁻¹ with increasing in LAA

concentration, since the interaction between GO sheets increase with higher reduction degree and the restacking of the 2D sheets results in smaller surface area. GO as 2D nanofluidic material with negative surface charge has the enhanced diffusion related to the strong ion-ion correlations under severe nanoconfinement. In this case, charge is transported through the pore by the flux of cations that interact with the immobile negative surface charges from carboxyl groups etc. Osmotic energy conversion could be boosted by such 3D aerogel interface, when pore diameter of around 2 nm or less is in the order of Debye screening length in the 2D nanofluidic channels. The Poisson–Nernst–Planck (PNP) model[50] was used to simulate the osmotic potential from the ion gradient of the 2D nanofluidic material and it also showed that the appropriate pore size for membrane-scale osmotic energy conversion should be around 2 nm, matching our experimental results.

The output performance of the power sources with different GO aerogels was evaluated by the potentiostatic linear sweep voltammetry with RTIL electrolyte. The GO aerogel with pore size of 2.1 nm made by GO:LAA ratio of 1:0.5 (Fig. 3f) shows the maximum $V_{OC}$ of 0.679 V and

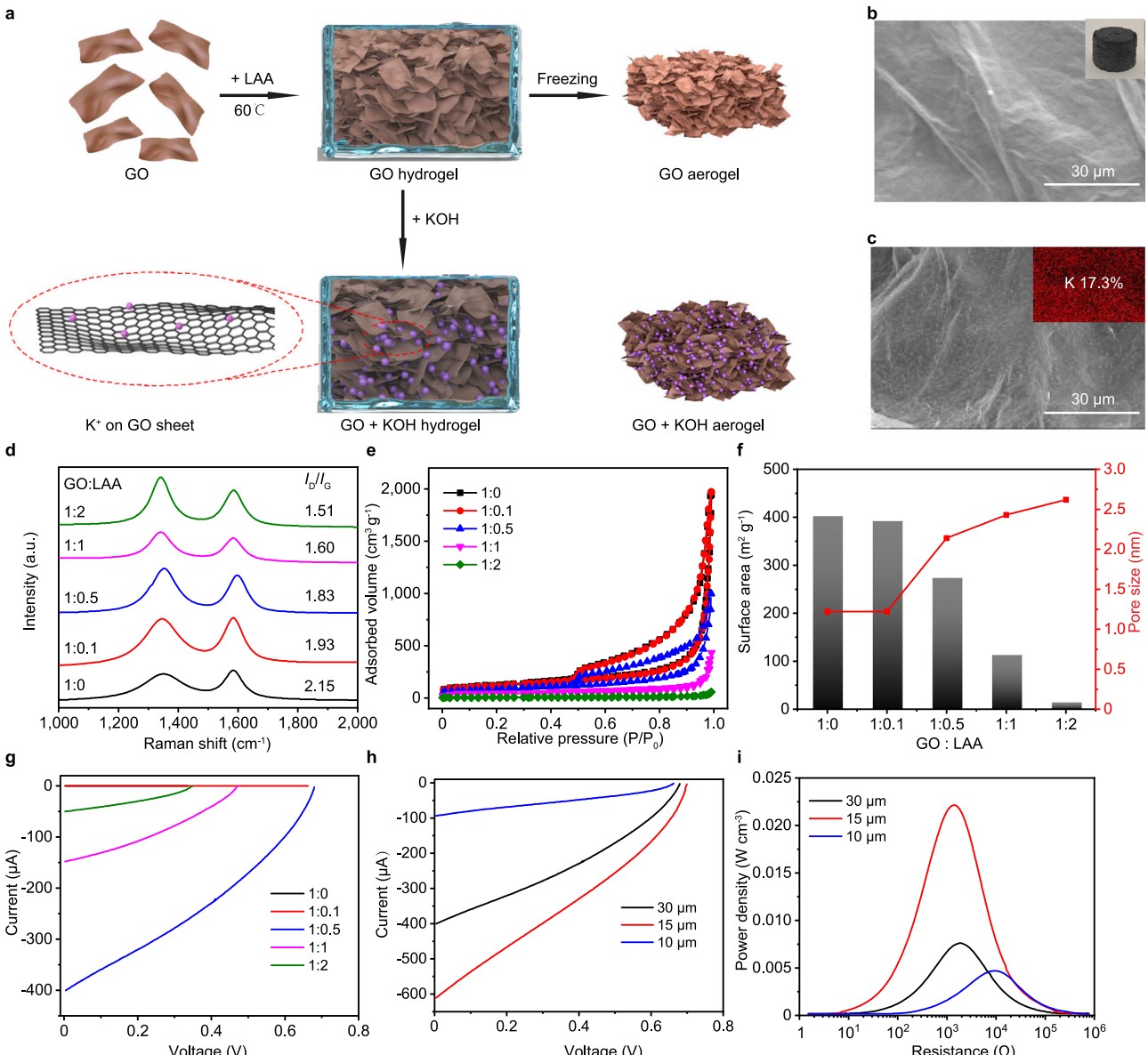

**Fig. 3 | GO aerogel preparation process and parameters for optimized osmotic power source in 3D. a** Schematic illustration of preparation process. The GO sheets were reduced by LAA at 60 °C forming a GO hydrogel and then dried through a freeze-drying process. For the GO + KOH aerogel, GO hydrogels were soaked in KOH aqueous solutions to introduce ion concentration difference before freezing. **b** SEM image of the GO aerogel. Inset photo of the obtained GO aerogel. **c** SEM image of the GO + KOH aerogel. Inset is the EDS element mapping of the GO + KOH aerogel. **d–f** The Raman spectroscopy (**d**), BET tests (**e**), surface area and pore size (**f**) of GO aerogels with different GO:LAA ratio. **g, h** I–V characteristics of power sources with GO aerogels that have different GO:LAA ratio (**g**) and thickness (**h**). **i** Power density of the power sources with different GO aerogel thickness. Source data are provided as a Source Data file.

$I_{SC}$ of 402 µA (Fig. 3g). The I−V characteristics of GO:LAA with ratio of 1:0 overlaps that of 1:0.1 in Fig. 3g, and the electrical properties of GO aerogel have little change with adding little amount of LAA. The four-point resistance measurement shows that the resistivity of the aerogels decreases dramatically with increasing LAA concentration (Supplementary Table 2). Besides the optimized size effect in the nanoconfinement, the maximum power of the device might be due to the balance of lower internal resistance with the increase in LAA concentration and higher ion storage capacity with larger surface area (Supplementary Table 2). It should be noticed that inset in Fig. 3b is the image of the as-made GO aerogel. When the GO aerogel was used in the osmotic power device, it was pressed forming nanopores (Supplementary Fig. 8), and the thickness of the aerogels also influences the output performance of the power source. As shown in Fig. 3h, although the $V_{OC}$ remains the same for aerogels with same LAA concentration under different thickness of 30, 15, and 10 µm, the aerogels with the

thickness of 15 µm has the highest $I_{SC}$ of 613 µA. The maximum output power density is calculated to be 22 mW cm⁻³ (Fig. 3i), which is similar to that of the ultrathin planar osmotic power source (28 mW cm⁻³) and 8 times higher than that of the ink-jet printed moisture-enabled power source (2.5 mW cm⁻³) reported before[40]. The improved electrochemical performance might come from the large surface area of the 3D aerogel architecture and the superior ionic conductivity of ionogel electrolyte. The optimized LAA concentration and aerogel thickness were used in the following study. To study the effects of the K⁺ concentration, GO aerogels were soaked in KOH solutions with different concentrations during the preparing process. As shown in Supplementary Fig. 7b, with the increasing in KOH concentration, the surface of the aerogels showed increasing number of small particles, indicating larger salt intake. These GO + KOH aerogels were paired with GO aerogels without KOH to test the $V_{OC}$ of the power sources (Supplementary Fig. 7c). With increase in KOH concentration, the K⁺ content

raises from 0% to 17.3% and the $V_{OC}$ increases from 0.01 V to 0.69 V. However, further increase in the KOH concentration could lead to crystallization and unevenly distributed cations, affecting the stability and reproducibility of the power sources. These results further suggest that the voltage of the power source mainly origins from the ion gradient between the two electrodes ($E_{diff}$), which is consistent with our previous results and other reports in power sources utilizing the ion gradient[40,51]. This also means that by selecting aerogels with different cation concentrations as building blocks, the voltage output could be customized based on the energy requirement. The directional ion migration needs to be converted to electron transportation to power external circuits. This electrochemical process is usually based on the chemical redox reactions or the physical adsorption of ions on the electrode surface. CV of such devices with and without RTIL in Supplementary Fig. 7d exhibit typical electric double layer capacitor (EDLC) behavior and no obvious redox peaks within the voltage window. RTIL enhances charging dynamics under severe nanoconfinement and accelerates ionic conductivity in the nanopores avoiding the humidity limitation for the osmotic power source and could even further overcome temperature limitations. Similar to the planar osmotic power source, the $I–V$ characteristics show that the $I_{SC}$ increases with increasing ionic conductivity while the $V_{OC}$ remains almost the same (Supplementary Fig. 7e) It should be noted that $I_{SC}$ is much higher than that of the power sources without RTIL, suggesting that RTIL electrolytes could significantly improve the output performance of GO aerogel the power sources as well. The CV curves and galvanostatic charge–discharge results (Supplementary Fig. 9a, b) reconfirm EDLC behavior of the power source based on GO aerogel. To eliminate the effect of the charge collectors, Ag, Au, copper (Cu), and carbon were all tested as conductive substrates and the CV curves showed similar shapes (Supplementary Fig. 9c). It should be noted that the CV of the devices without RTIL shows capacitive behavior with various types of charge collectors, different from the previous planar osmotic power sources and moisture-based power sources with redox reactions[40]. This might be due to the large amount of ion adsorptions on porous aerogel surfaces, which dominates the electrochemical process. $V_{OC}$ equals to $E_{diff}$ in the osmotic power source made of nanoporous GO aerogels and the current purely comes from the ion gradient.

## Self-healing ionogels

The challenge of application of RTIL in devices is related to its "liquid" nature. The ionogel electrolyte consisting of RTIL (TESTFSI) with poly(vinylidene fluoride-co-hexafluoropropylene) (PVDF-HFP) was used here. It is reported that the cations in the ionic liquid and the PVDF-HFP polymer chain in the fluoro-elastomers could interact via ion-dipole interactions, giving rise to self-healing capability[52,53]. This self-healing ionogel with tunable ionic conductivity and antifreeze properties could improve the output performance and allow versatile design strategies of the power sources for portable devices. The ionic conductivity of the ionogels was also evaluated by the EIS (Supplementary Fig. 10a, b) and the conductivity increases by three orders of magnitude from $10^{-6}$ S cm$^{-1}$ to $10^{-3}$ S cm$^{-1}$ as the RTIL content raised from 30 to 70 wt.%. By tuning the mass ratio of the RTIL and polymer matrix, the ionic conductivity of the electrolyte could be customized on demands. The Tg of the ionogels with 70 wt.% RTIL obtained from the differential scanning calorimetry (DSC) test is −72 °C, significantly lower than −19.7 °C for the pure PVDF-HFP polymer (Supplementary Fig. 10c). This could be attributed to the ion-dipole interaction of the RTIL and the polymer chain. The RTIL could act as a plasticizer weakening the interactions between host polymer chains, therefore increase the flexibility and processability of the ionogels[54]. The mobile polymer chains could also facilitate the self-healing and increase the ionic conductivity[53]. The ionic conductivity of the ionogels with 70 wt.% RTIL increases

with the increasing temperature from −40 to 50 °C (Supplementary Fig. 10d, e).

Such ionogel could be cut and then placed together, exhibiting fast self-healing capability at ambient environment shown in Supplementary Fig. 11a. The tensile experiments were performed by an Instron E3000 instrument at 25 °C at a strain rate of 5 mm min$^{-1}$. Considering the restoration of both stress and strain, the mechanical self-healing efficiency was defined as the proportion of toughness restored relative to the original toughness[55]. As shown in Supplementary Fig. 11b, the self-healing ionogel shows characteristic feature of the stress-strain curve of self-healing ionogels reported by other groups[52,53]. Compared with the pristine ionogels before being cut, the healed ionogels maintained similar Young's modulus and tensile stress curves (Supplementary Fig. 11b, c). The ionic conductivity of the ionogel was evaluated by EIS (Supplementary Fig. 11d) and indicated that there was also no obvious difference for the ionic conductivity of ionogel before and after being cut (Supplementary Fig. 11e), which remained in the same orders of magnitude ($10^{-3}$ S cm$^{-1}$). Such effective self-healing capability might be attributed to the highly reversible ion-dipole interactions of the ionogel[53].

## Modular design and scalable integration of the osmotic power source

A modular design concept could be demonstrated (Fig. 4a), in which the output performance of the osmotic power sources could be tailored on demands by assembling building blocks with customizable properties. GO aerogels with adjustable cation concentrations are selected based on the voltage requirements and ionogels with programmable conductivities could be drop-casted onto aerogels affecting the current output. The component parts are stored separately before use, avoiding self-discharge. When needed, GO aerogel electrodes could be adhered together by ionogels through the self-healing process without external stimulation. As shown in Fig. 4b, the power source with optimized GO aerogels and ionogels with 70 wt.% RTIL has the $V_{OC}$ of 0.6 V and the $I_{SC}$ of 440 μA with the maximum volumetric specific power density of 12 mW cm$^{-3}$. The modular design could enable the construction of custom-built power sources with prefabricated components and meet the energy requirements of various integrated electronic systems. Proof of concept of the plug-unplug modular design in the osmotic power source was demonstrated by comparing the output power for the self-healing ionogel before and after being cut. As shown in Supplementary Fig. 12a, the modular design power source with the ionogel after healing has the $V_{OC}$ of 0.62 V and the $I_{SC}$ of 410 μA with its power density still around 11.8 mW cm$^{-3}$ vs. 12.0 mW cm$^{-3}$ (Fig. 4b) from the pristine device. Thus, the power generation of the modular design power source is not affected a lot after healing. For the long-term performance, further experiments were carried out using the component parts such as the cathode GO aerogels, ionogels and anode rGO aerogels, which had been kept separately in the glovebox for half year. The cathode GO aerogel electrode and anode rGO aerogels were adhered together by ionogels through the self-healing process. Similar $V_{oc}$ and $I_{sc}$ are obtained and the volumetric specific power density is around 11.1 mW cm$^{-3}$ (Supplementary Fig. 12b), showing perfect long-term performance. By adopting the ionogels as electrolyte, the iontronic power source could also operate in harsh environments like low humidity and subzero temperature. As shown in Fig. 4c, the assembled power source exhibit $V_{OC}$ of 0.54 V and the $I_{SC}$ of 57 μA at −10 °C under nearly 0% relative humidity. With raising temperature, the $I_{SC}$ increases due to the enhanced ionic conductivity, while the $V_{OC}$ remains almost the same (Fig. 4d). In contrast, the moisture-based power source without RTIL electrolyte has low current output and could not work at temperature below zero (Supplementary Fig. 13), while this power source has much larger current output and could work even at the temperature of −40 °C, greatly broadening the application conditions.

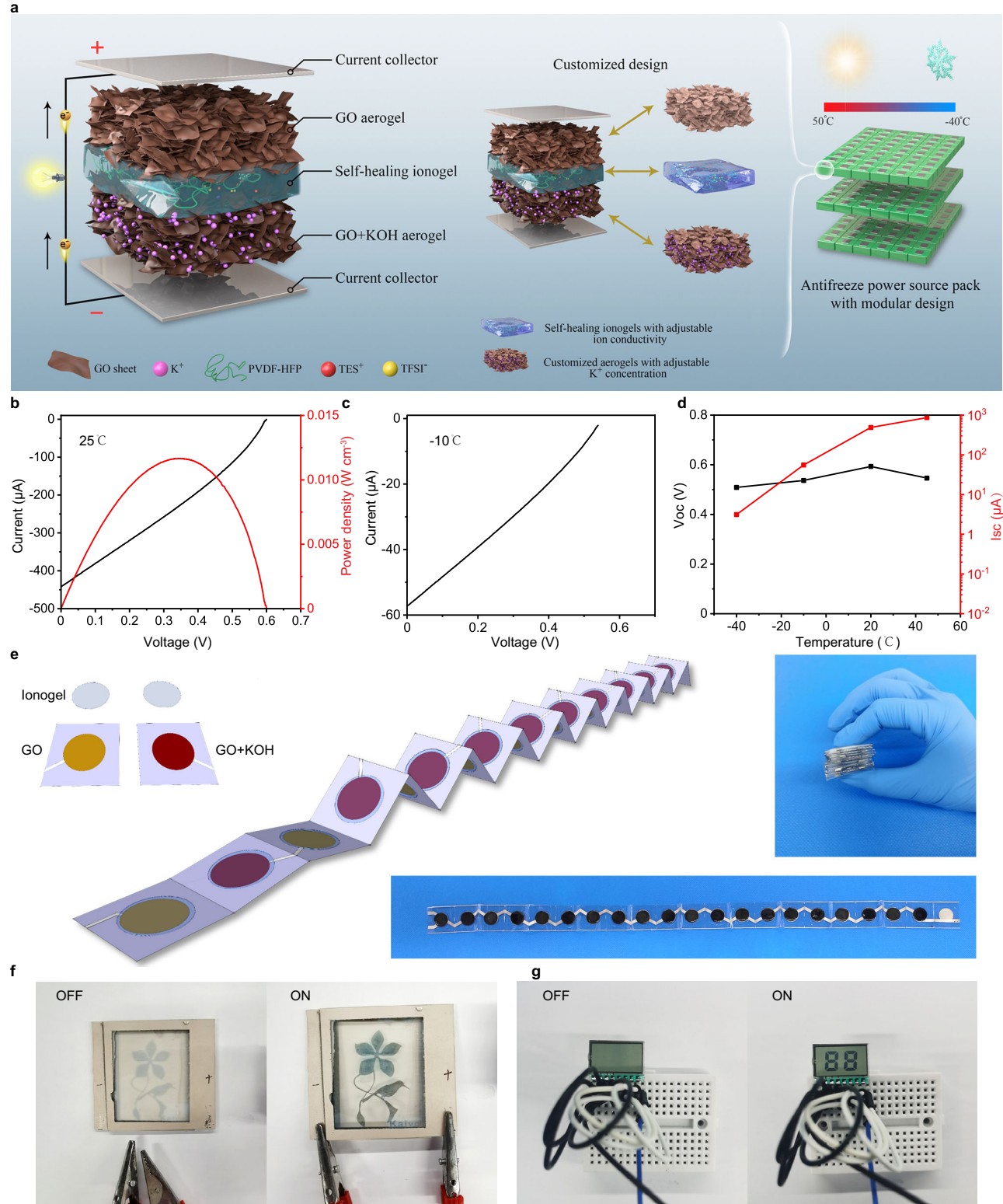

**Fig. 4 | Modular design osmotic power source. a** Schematic illustration of the modular design device construction. Two GO aerogels with different K$^+$ concentrations are sandwiched between charge collectors and separated by an ionogel that consists of RTIL (TESTFSI) and PVDF-HFP polymer matrix. **b** I–V characteristic and power density of the GO aerogel-based power source with ionogel electrolyte at 25 °C. **c** I–V characteristic of the power source at −10 °C. **d** The output performance of the power source at different temperatures. **e** The foldable design of 20 devices in series connection on both sides of the substrate enabled by self-healing ionogels. **f, g** The foldable power source could power the electrochromic device (**f**) and the liquid crystal display screen (**g**). Source data are provided as a Source Data file.

By manipulating different components of such type of power source, it offers a handy assembling process and abundant potential designs, for example, a simple Origami design of foldable power source was fabricated by connecting 20 devices in series on both sides of the film substrate (Fig. 4e and Supplementary Fig. 14a). A repeating series of device components on the flat substrate were stacked together forming a sandwich structure by taking advantage of the folding strategy. The tiny bulk of the folded osmotic power source from pure ion gradient has the $V_{OC}$ of approximate 10 V and can power not only electrochromic devices (Fig. 4f, Supplementary Fig. 14b and Supplementary Movie 2) but also the liquid crystal display screen (Fig. 4g and Supplementary Fig. 14c), providing a versatile design for powering various electronic devices by osmotic energy from just ion gradient. Such modular design power source offers the areal power density of 1.3 mW cm$^{-2}$ at ambient environment that is significantly higher than 0.5 mW cm$^{-2}$, which has been flagged as the target for making salinity gradient power economically viable[2]. The outputs (in mW cm$^{-2}$) of various osmotic power sources with different materials were compared in Supplementary Table 3, and our modular design power source generates by far the highest areal power density among them.

## Discussion

GO is an ideal candidate to fabricate iontronics due to the surface charge and the 2D nanofluidic channels could be readily formed from GO when restacked. Such nanoscale confinement is very similar to the transport of ions passing through nanometer-sized biological channels accounting for a wide array of physiological processes. The asymmetric charge distribution of GO/rGO and ion transport inside GO lead to the osmotic energy. Remarkably, the osmotic power source reported in this paper yielded ionic currents in the microampere regime, which are orders of magnitude higher than those reported previously. The volumetric specific energy density (6 mWh cm$^{-3}$) of the planar osmotic power source is comparable to thin-film lithium batteries and the power density (28 mW cm$^{-3}$) is as high as supercapacitors. Such printable planar power source could be conveniently integrated into other electric circuit and/or devices. Self-charging triboiontronic device with total thickness less than 200 μm was shown by coupling energy harvesting TENG with ultrathin GO osmotic power source, which was conformal to various irregular surface. Generally speaking, the voltage from the GO osmotic power source mainly comes from K$^+$ concentration gradient ($E_{diff}$) between rGO and GO. The output current can be enhanced either by confined electrochemical redox reactions between GO and Au charge collector in the planar power source, or by amplifying ion current through incorporating nanoporous GO aerogels and self-healing ionogels with programmable conductivities in 3D. The osmotic power source based on GO aerogels with nanoconfined structure (pore size ~2 nm) can convert the chemical potential energy of the ion gradient to electric energy through ion transportation driven purely by the ion gradient. RTIL accelerates the kinetics of ion diffusion in nanoconfined channels and it also imparts the osmotic power source to overcome the operation limitation on humidity and temperature. Benefiting from the antifreeze RTIL ionogels, the power source could operate even at the temperature of −40 °C, bringing it further step towards practical applications. The output performance of such power source could be tailored based on demands of the electronic systems by selecting pre-fabricated building blocks like Lego. An Origami inspired foldable power source was constructed demonstrating the handy assembling process and abundant potential designs for powering electronic devices by ionic power. Modular design power source offers areal power density of 1.3 mW cm$^{-2}$ at ambient environment making salinity gradient power economically viable. This work advances the fundamental understanding of function of 2D nanofluidic material in iontronics and the iontronics based on GO has distinct advantages such as low cost, safe, ease of scaling up to

support high ionic currents and flexibility. By combining 2D nanofluidic material in such ion gradient system, it also opens up abundant potential designs for development of ionic diode/transistor, ionic circuits and photo-induced iontronics etc. The iontronic devices possess the ability to regulate ion flow in direction and magnitude, and they especially could find applications in the development of wearable or implantable iontronic devices or even neuronal-computer interfaces in the future.

## Methods

Our research complies with all relevant ethical regulations, overseen by Beijing Institute of Nanoenergy and Nanosystems, Chinese Academy of Sciences.

### Materials

GO was prepared from graphite powders (XFNANO, INC) using a modified Hummers method[56]. Graphite powders (1 g), H$_3$PO$_4$ (2 mL) and H$_2$SO$_4$ (21 mL) were stirred together for 2 h in an ice bath, and KMnO$_4$ (3 g) was added slowly. The solution was then transferred to a 35 °C water bath and stirred for ca. 0.5 h to form a thick paste. Deionized water (200 ml) and H$_2$O$_2$ (10 mL) were added successively at 95 °C, and the color of solution turned to bright yellow from dark brown. After several centrifugations to wash away soluble salts and incompletely reacted graphite. The brown-yellow clarified liquid collected is the graphene oxide dispersion and the upper liquid was freeze-dried to form GO powder. All related chemicals of H$_2$SO$_4$, KMnO$_4$, H$_2$O$_2$, H$_3$PO$_4$ etc. were purchased from Sigma-Aldrich and were used as received. The GO solution (5 mg mL$^{-1}$) was prepared by dissolving the GO with deionized water. The rGO solution was prepared by adding 0.1 mol L$^{-1}$ KOH into the GO solution (5 mg mL$^{-1}$) with a 1:2 (vol/vol) ratio. KOH, acetone, LAA, BMIMTFSI, TESTFSI and EMIMBF$_4$ etc. were purchased from Sigma-Aldrich. PVDF-HFP was purchased from 3 M Company. Deionized (DI) water, having a resistivity of above 18 MΩ cm$^{-1}$ was collected from a Mili-Q Biocel system. The target material Au (99.9999%) was purchased from ZhongNuo Advanced Material (Beijing) Technology Co., Ltd.

### Fabrication of flexible Au charge collector

Commercial polyethylene terephthalate (PET) film was thoroughly cleaned by abundant acetone, alcohol and deionized water with a bath sonicator. A computer-controlled commercial CO$_2$ laser cutter system was used to cut the PET film into predesigned patterns as the shadow mask, and then it was pasted on the no patterned PET film with the Kapon both side tape. Au was then deposited on the exposed PET substrate with the aid of the mask via the magnetron sputtering film deposition system (Denton Discovery 635). GO and then rGO solutions were deposited onto PET substrate with sputtered gold strips as charge collectors.

### GO aerogel and PVDF-HFP ionogel preparation

GO aerogel was fabricated through a self-assembled chemical reduction and freeze-drying process. The GO sheets with abundant functional groups could be partially reduced forming a GO hydrogel in a glass vial under a mild condition in the presence of leukocyte ascorbic acid (LAA)[49]. GO aerogel was then obtained through a freeze-drying process to remove the excessive water while maintaining the ordered interconnected network with limited sheet restacking or structure collapsing. For the GO + KOH aerogel, GO hydrogels were immersed into different concentrations of KOH aqueous solution (0.005, 0.01, 0.05, 0.1 M) at room temperature for overnight before freeze-drying. To form the ionogel film, PVDF-HFP was dissolved by acetone and mixed uniformly with TESTFSI (0, 30, 50, 70, 100 wt.%) at room temperature. After obtained through a casting and drying process at 75 °C for 48 h in a vacuum oven, the ionogel film was prepared.

## Modular design power source construction

The prepared GO aerogels and GO + KOH aerogels were pressed with 10 MPa for 10 min and cut into circular electrodes with the diameter of 16 mm. For mechanism study, GO aerogel electrodes and GO + KOH aerogel electrodes were clamped between two conductive charge collectors on polyethylene glycol terephthalate substrates and separated by the cellulose separator soaked with RTIL electrolyte to avoid direct contact. For foldable power source, firstly the electrodes were put onto the substrate with screen-printed silver patterns. Then the RTIL and polymer mixture were drop-casted on top of the electrodes and subsequently dried in the vacuum oven at 55 °C for 2 days to remove acetone solvent.

## Characterization and measurements

Morphologies of devices were observed by the SEM (SU8020, Hitachi) with the 5.0 kV accelerating voltage, 10 μA emission current and EDS was used for the element analysis. AFM was performed at room temperature with an Asylum Research MFP-3D in tapping mode with a scanning rate of 0.2 Hz. Dektak XT™ stylus profiler (Brucker) was used to profile the morphology of the planar osmotic power sources and provides information on volume values for the calculation of specific energy and power density. All electrochemical measurement were carried out with the electrochemical workstation (Multi Autolab M204). Electrochemical impedance spectrum (EIS) was measured on an extended range from 1 MHz to 0.01 Hz. Temperature and relative humidity were controlled by an environmental simulation test chamber (Vötsch Technik). The BET measurement was carried out on a MicrotracBEL BELSORP-max instrument by $N_2$ physisorption at 77 K. Raman spectra were collected with the LabRAM HR Evolution (Horiba) with a 532 nm laser. Fourier-transform infrared (FTIR) spectra were collected by Bruker VERTEX80v.The resistivity and the thickness of the aerogels were tested by the four-point probe method and the digimatic indicator (Absolute 543-490B, Mitutoyo Corp.), respectively. The DSC measurement were performed using the DSC Q2000 from −90 °C to 30 °C with the speed of 5 °C min⁻¹. For the TENG characterization, A step motor (LinMot E1100) was used to provide the input of mechanical motions. The voltage and current output were recorded by a Keithley electrometer 6514. The tensile experiments were performed by an Instron E3000 instrument. Zeta potential were obtained by using a Beckman coulter instrument (Delsamax Pro).

## Reporting summary

Further information on research design is available in the Nature Research Reporting Summary linked to this article.

## Data availability

The authors declare that all the data that support the findings of this study are available within the article and its supplementary information files. Source data are provided with this paper.

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

## Acknowledgements

We thank P.G. Peng from Xiangtan University for drawing some illustration pictures. The authors appreciate the technical assistance from the instrument & equipment platform of Beijing Institute of Nanoenergy and Nanosystems.

## Author contributions

D.W. and Z.L.W. proposed the idea and the project. D.W. and F.Y.Y. designed the experiment and performed the device fabrication and characterization. Z.H.J. performed the TENG fabrication and characterization. D.W. and Z.L.W. supervised the project. All the authors discussed the results and commented on the manuscript.

## Competing interests

The authors declare no competing interests.
