## [Peer Review File · Nature Communications]

Flexible iontronics based on 2D nanofluidic materialREVIEWER COMMENTS

Reviewer #1 (Remarks to the Author):

This manuscript titled in "Flexible iontronics based on 2D nanofluidic material" demonstrates an ultrathin osmotic power source with voltage of 1.5 V, volumetric specific energy density of 6 mWh cm⁻³ and power density of 28 mW cm⁻³, which can form a self-charged conformable triboiontronic device to be operated under -40 °C and overcome humidity limitations. This work is original and reasonably well executed. I believe the work will be of interest to many in materials science. Before considering this manuscript for publication, the authors should consider the following points in any revision as follows:

1. What about the performance of thin TENG film on the skin in humid environment?
2. How about the output of 3D osmotic power source compared with other source?
3. What is the relationship between pore size of gels and ion transport in the system?
4. In a sweep voltage range of -2 V~2 V, there will occur electrolytic water reactions. As we can see, the current signal is non-linear with voltage change in the absence of AgNO₃.
5. In Figure 1a, what is the structural formula for rGO? In Figure 3g, the data of 1:0 overlaps the data of 1:0.1?
6. In Supplementary Figure 2a and b, the y-coordinate should be shortened.
7. In the figure, the spaces are required between the number and the unit, such as in Figure 1b, 1c, Figure 3b and 3c.
8. In Figure 3b and 3c, the SEM images of the GO layer and rGO (cross section) should be a unified scale for the observation of graphene layered structure. The scale bar in Supplementary Figure 3 a and b should be added.
9. In Supplementary Figure 2g, the EIS data of the osmotic power sources without RTIL is confused, and we can't get the conclusion about the charge transfer resistance.
10. The authors mentioned, the structure of GO aerogel is nanopores with diameter of several nanometers, but in Figure 3b, there are pores with diameter of hundreds of micrometers.
11. For the EIS data, the equivalent circuit diagram should be fitted first, and then to compare the impedance value. So, the EIS part in the manuscript should be revised.
12. Whether the Voc of the device can be further increased by changing the redox potential by other electrochemical reaction?
13. Many recent publications related to nanofluidic iontronics have been published, such as Science, 2021, 373: 628-629, Chinese Chemical Letters, 2021, 32: 642-648, Nano Today, 2020, 33: 100868, Nano Letters, 2020, 20, 6937-6946, Advanced Materials, 2019, 31: 1805130, et al. In order to help readers better understand the important of this work, these references should be cited, correspondingly.

I will be happy to recommend for publication a revised version of the manuscript in Nature Communications.

Reviewer #2 (Remarks to the Author):

This is an interesting paper related with flexible iontronics based on nanofluidic materials. They developed novel foldable power source by using iontronics. The work could be publishable in Nature Communications with the following minor correction.

- 1) Lego-like osmotic power source: The reviewer thinks the word "Stacked" is better than the word "Logo-like".
- 2) Stability of the device needs to be mentioned.
- 3) Conductivity of rGO needs to be provided.
- 4) It is better to cite the following references for graphene iontronic pressure sensor.
 - Kim et al., Adv. Funct. Mater. 30, 2070089(2020)
 - Nie et al., Adv. Mater. 27, 6055 (2015)

Reviewer #3 (Remarks to the Author):

This manuscript of "Flexible iontronics 1 based on 2D nanofluidic material" reports on a GO-based iontronic device that can harvest energy with a high power density compared to those have already been published. The performance of energy conversion can be boosted by incorporating: asymmetric distribution of surface charges of GO/rGO, 3D gel structure, and the addition of RTILs. I like the Lego-like design that consists of GO aerogels and GO+KOH layers sandwiching a self-healing gel. The ultrathin and wearable features make the real application possible. The device can even work in harsh environments. This manuscript is informative and should be of interest to a broad audience. I'm willing to support its publication after the following comments have been addressed.

1. The authors adopt previous work to claim 2 nm is optimal for energy conversion. However, continuum-based theory breaks down for confinement less than ~ 3 nm and may become invalid for the system with RTILs. This explanation may be OK for this paper but a brief justification is necessary.
2. The authors propose surface charge is essential for osmotic energy. While Fig. S1a shows a surface-charge-governed feature, it is difficult to determine the charge density of porous membranes through ionic conductance measurement. The plateau may result from other factor such as contaminations. Did authors measure the zeta potential of the GO membrane to confirm its charge property? Besides, Fig. S1 was performed with KCl solution rather than the solution used in the main text. Did authors observe the similar nonlinear conductance under the experimental condition, for example, KOH with RTIL?
3. How do the authors obtain the ion selectivity of 0.99 in Supplementary Discussion? It seems the values for activities are not provided. It is important to clarify the calculation of activity. Ionic activity in ionic liquids could be complicated because of the significant short-range correlation. In that case Debye-Huckel theory fails to describe enhancement of activity at high concentrations.
4. (following above) 0.99 means a nearly perfect ion selectivity. It's puzzling because the surface charge density of GO membrane was reported as ~ 2 mC/m² (Zhang et al., Nat Commun 10, 1253 (2019)). This indicates that the electrostatic repulsion is not such strong. Is there other mechanism, for example, steric exclusion or interactions between RTIL and GO surface?
5. Line 296, a power density of 1.3 mW cm⁻² was obtained from Lego-like power source and was compared to commercialization benchmark of 0.5 mW cm⁻². Is the tested surface area 0.6 cm \times 0.2 cm, according to Supporting Information? If so, this number is still below realistic conditions of commercial ion-exchange membranes. Since the power density is highly dependent on the membrane area (Xiao et al., Joule 3, 2364–2380, 2019), the tested area should be specified when making a comparison.
6. Self-healing materials have been used to make advanced membranes in water filtration. Fig. S6 demonstrates the self-healing property for ionogels but I'm curious whether the authors compare the output power before and after the cut. Would the power generation be affected a lot after healing? Also, it would be better if typical self-healing test (i.e., stress-strain curve) can be provided. Longer-term performance of self-healed power generator would significantly elevate this work.
7. Here are several minors :
 - Improve the resolution of Fig 1f
 - Line 46, it states that power densities of RED are small, but the cited ref 21,26 are harvesting energy through RED. Generally, "surface" power density (e.g., W/m²) is commonly used in RED power generation.

We are deeply appreciated to all reviewers' helpful comments. Provided below is our detailed response to each comment/suggestion. The specific changes made to the manuscript to address each point are highlighted in yellow.

Responses to Reviewers' Comments:

Reviewer #1 (Remarks to the Author):

This manuscript titled in “Flexible iontronics based on 2D nanofluidic material” demonstrates an ultrathin osmotic power source with voltage of 1.5 V, volumetric specific energy density of 6 mWh cm⁻³ and power density of 28 mW cm⁻³, which can form a self-charged conformable triboiontronic device to be operated under -40 °C and overcome humidity limitations. This work is original and reasonably well executed. I believe the work will be of interest to many in materials science. Before considering this manuscript for publication, the authors should consider the following points in any revision as follows:

Response: We are grateful to the reviewer for the thorough review and help to improve the paper. We are appreciated that the reviewer thought our work is of interest to many in materials science.

1. What about the performance of thin TENG film on the skin in humid environment?

Response: The open-circuit voltage (Voc) and short-circuit current (Isc) under different relative humidity (RH) conditions were tested for the thin TENG film pasted on the skin (Supplementary Fig. 6).

All experiments were carried out at 25 °C under four different RH conditions. The Voc and Isc of the TENG remained relatively stable, around 20 V and 0.35 μA, under RH of 20%, 40%, and 60% respectively. Only under very high humidity conditions (RH of 80%), the Voc and Isc of the TENG decreased slightly to 12 V and 0.18 μA, respectively.

Experiments have shown that the output performance of the TENG is quite stable under ambient temperature and humidity conditions. Only in the case of very high humidity conditions (RH higher than 80%), the output performance of the TENG will be slightly reduced. This proves that when the TENG is pasted on human skin, it can operate normally and the output performance is relatively stable. It should also be noted that in this experiment, the TENG was driven by flapping the human skin to generate output electrical signals in the environmental simulation chamber (Vötsch Technik), and the impact flapping force exerted by the human hand on the TENG is not as identical as that exerted by the linear motor, resulting in the little difference in the output performance reported in Fig. 2c and 2d.

The corresponding discussion has been added in the revised manuscript. Detailed revision is as below.

In the revised manuscript (Page 10-11, line 219-225):

“Furthermore, the Voc and Isc under different humidity conditions, RH of 20%, 40%, 60%, 80% were tested for the thin TENG film pasted on the skin as shown in Supplementary Fig. 6, respectively. It was shown that the output performance of the TENG is quite stable under ambient temperature and humidity conditions. Only in the case of very high humidity conditions (RH higher than 80%), the output performance of the TENG will be slightly reduced. This proves that when the TENG is pasted on human skin, it can operate normally and the output performance is stable.”

Supplementary Fig. 6 Output performance of the TENG pasted on skin under different relative humidity conditions. a, open-circuit voltage. b, short-circuit current.

The TENG was driven by flapping the human skin to generate output electrical signals in the environmental simulation chamber (Vötsch Technik), and the impact flapping force exerted by the human hand on the TENG is not as identical as that exerted by the linear motor, resulting in the little difference in the output performance reported in Fig. 2c and 2d.

2. How about the output of 3D osmotic power source compared with other source?

Response: To make the comparison of the 3D osmotic power source, the outputs (in mW cm^{-2}) of various osmotic power sources with different materials were listed in Supplementary Table 2. Our 3D modular design power source generates by far the highest areal power density among them. If compared in volumetric power density, the 3D osmotic power source generates 12 mW cm^{-3} as high as supercapacitors shown in the Ragone plot in Fig.1f.

The corresponding discussion has been added in the revised manuscript. Detailed revision is as below.

In the revised manuscript (Page 18, line 377-379):

“The outputs (in mW cm^{-2}) of various osmotic power sources with different materials were compared in Supplementary Table 2, and our modular design power source generates by far the highest areal power density among them.”

Supplementary Table 2 Comparison of the modular design osmotic power source output with various materials.

Type	Materials	Power density (mW cm^{-2})	Reference
This work	GO	1.3	
2D materials	GO	0.526	Ref. 2
	GO	0.077	Ref. 3
	GO	0.035	Ref. 4
	rGO	0.115	Ref. 5
	Mxene	0.21	Ref. 6
	Mxene/Kevlar	0.41	Ref. 7
Polymer	PES-Py/PAEK-HS	0.266	Ref. 8
	PMA/PS-b-P4VP	0.38	Ref. 9
	Polyimide	0.026	Ref. 10

Others	Silicon	0.77	Ref. 11
	SNF/AAO	0.286	Ref. 12
	Hydrogel/ANF	0.506	Ref. 13
	Mesoporous carbon/AAO	0.346	Ref. 14

3. What is the relationship between pore size of gels and ion transport in the system?

Response: Basically, the pore size of the aerogel was controlled by the LAA concentration as shown in Supplementary Table 1. Both the resistivity of the GO aerogel and its surface area decreased with increase in LAA concentration, and this may be due to that the interaction between GO sheets increases with higher reduction degree and the restacking of the 2D sheets results in smaller surface area. On the other hand, *charge is transported through the pore by the flux of cations that interact with the immobile negative surface charges from carboxyl groups etc. Osmotic energy conversion could be boosted by such 3D aerogel interface, when pore diameter of around 2 nm or less is in the order of Debye screening length in the 2D nanofluidic channels.* The Poisson-Nernst-Planck (PNP) model¹ was used to simulate the osmotic potential from the ion gradient of the 2D nanofluidic material and it also showed that the appropriate pore size for membrane-scale osmotic energy conversion should be around 2 nm, matching our experimental results.

Besides the optimized size effect in the nanoconfinement, the maximum power of the device might be due to the balance of lower internal resistance with the increase in LAA concentration and higher ion storage capacity with larger surface area (Supplementary Table 1).

The corresponding discussion has been added in the revised manuscript. Detailed revision is as below.

In the revised manuscript (Page 12-13, line 252-258 & 265-268):

“In this case, charge is transported through the pore by the flux of cations that interact with the immobile negative surface charges from carboxyl groups etc. Osmotic energy conversion could be boosted by such 3D aerogel interface, when pore diameter of around 2 nm or less is in the order of Debye screening length in the 2D nanofluidic

channels. The Poisson-Nernst-Planck (PNP) model⁵⁰ was used to simulate the osmotic potential from the ion gradient of the 2D nanofluidic material and it also showed that the appropriate pore size for membrane-scale osmotic energy conversion should be around 2 nm, matching our experimental results.”

“Besides the optimized size effect in the nanoconfinement, the maximum power of the device might be due to the balance of lower internal resistance with the increase in LAA concentration and higher ion storage capacity with larger surface area (Supplementary Table 1).”

4. In a sweep voltage range of -2 V~2 V, there will occur electrolytic water reactions. As we can see, the current signal is non-linear with voltage change in the absence of AgNO₃.

Response: To investigate the possibility of the electrolytic water reactions, we carried out cyclic voltammetry test for the osmotic power sources with AgNO₃ (Au/AgNO₃/GO/RTIL/rGO/Au) and without AgNO₃ (Au/GO/RTIL/rGO/Au). It was shown that there is no Faradaic currents coming from electrolysis of water in a sweep voltage range between -2 V~2 V (Supplementary Fig. 4f).

Although the theoretical voltage of electrolysis of water is 1.23 V, the practical voltage for water electrolysis is generally over 1.7 V and is influenced by many parameters such as pH, salt concentration etc. We did not observe electrolysis of water in the sweep voltage range between -2 V~2 V in Supplementary Fig. 4f. *The non-linear I-V characteristic of the device without AgNO₃ in Supplementary Fig. 4e in the negative voltage range, may be due to the ionic rectification*, i.e. facilitation of K⁺ cation transport through negatively charged 2D nanofluidic channels of GO material under humidity. The ionic rectification from the 2D nanofluidic materials results in the osmotic energy and both devices don't contain liquid water. In addition, both devices in Supplementary Fig. 4e were covered with RTIL which has a large and stable electrochemical window.

The corresponding discussion has been added in the revised manuscript. Detailed

revision is as below.

In the revised manuscript (Page 6-7, line 132-142):

“Ionic rectification from the 2D nanofluidic materials is the origin of the osmotic energy. The non-linear I-V characteristic of the devices in Supplementary Fig. 4e in the negative voltage range, may be due to the ionic rectification, i.e. facilitation of K^+ cation transport through negatively charged 2D nanofluidic channels of GO material under humidity. To investigate the possibility of the electrolytic water reactions, we carried out cyclic voltammetry test for the osmotic power sources with $AgNO_3$ (Au/ $AgNO_3$ /GO/RTIL/rGO/Au) and without $AgNO_3$ (Au/GO/RTIL/rGO/Au). It was shown that there is no Faradaic current coming from electrolysis of water in a sweep voltage range between -2 V~2 V (Supplementary Fig. 4f) in both devices. Although the theoretical voltage of electrolysis of water is 1.23 V, the practical voltage for water electrolysis is generally much higher than that and is influenced by many parameters such as pH, salt concentration etc.”

Supplementary Fig. 4 f, Cyclic voltammograms of the osmotic power sources with $AgNO_3$ (Au/ $AgNO_3$ /GO/RTIL/rGO/Au) and without $AgNO_3$ (Au/GO/RTIL/rGO/Au) at 25°C at the scan rate of 10 mV s⁻¹.

5. In Figure 1a, what is the structural formula for rGO? In Figure 3g, the data of 1:0 overlaps the data of 1:0.1?

Response: The empirical structure of rGO with less functional groups containing oxygen was shown in Supplementary Fig. 1d.

FTIR (Fourier Transform Infrared Spectrometer) experiments were carried out to characterize the structure changes in GO after addition of potassium hydroxide (KOH), forming the rGO. FTIR spectrum (Supplementary Fig. 1a) shows that GO has the characteristic peak at 3402 cm^{-1} , which corresponds to the hydroxyl group (-OH). The peaks at 1726 , 1630 , 1386 , 1145 and 968 cm^{-1} correspond to the stretching vibration of C=O in the carboxyl group, C=C on the sp^2 carbon skeleton, C-OH group on the carbonyl group (-COOH), C-O-C group and C-O group on the epoxy group, respectively. Thus, GO has a large number of oxygen-containing functional groups in the empirical structural formula as shown in Supplementary Fig. 1b. rGO was prepared by mixing GO with KOH. It was reported that KOH could partially remove the oxygen-containing groups of GO sheets through a series of deoxygenation reactions^{2,3}. In Supplementary Fig. 1c, FTIR spectrum also shows a reduction in the amount of hydroxyl and carboxyl groups present in rGO. Therefore, the empirical structural formula of rGO contains fewer oxygen-containing functional groups as shown in Supplementary Fig. 1d.

The corresponding discussion has been added in the revised manuscript. Detailed revision is as below.

In the revised manuscript (Page 4, line 73-75):

“FTIR (Fourier Transform Infrared Spectrometer) characterization and the related empirical structural formula of GO and rGO were shown in Supplementary Fig. 1. A reduction in the amount of hydroxyl and carboxyl groups happens to the structure of GO after addition of KOH, forming the reduced GO (rGO).”

Supplementary Fig. 1 FTIR spectrum of GO and rGO with their empirical structural formula. a, FTIR spectrum of the GO. **b,** The empirical structural formula of GO. **c,** FTIR spectrum of the rGO. **d,** The empirical structural formula of rGO. FTIR spectrum of GO shows the characteristic peak at 3402 cm^{-1} , which corresponds to the hydroxyl group (-OH). The peaks at 1726 , 1630 , 1386 , 1145 and 968 cm^{-1} correspond to the stretching vibration of C=O in the carboxyl group, C=C on the sp^2 carbon skeleton, C-OH group on the carbonyl group (-COOH), C-O-C group and C-O group on the epoxy group, respectively. FTIR spectrum also shows a reduction in the amount of hydroxyl and carboxyl groups in rGO.

The data of 1:0 overlaps the data of 1:0.1 in Fig. 3g, and the ratio of 1:0 and 1:0.1 refers to the GO:LAA ratio. This means that with little amount of LAA, the electrical properties of GO aerogel have little change. The corresponding discussion has been added in the revised manuscript. Detailed revision is as below.

In the revised manuscript (Page 12-13, line 262-265):

“The I-V characteristics of GO:LAA with ratio of 1:0 overlaps that of 1:0.1 in Fig. 3g, and the electrical properties of GO aerogel have little change with adding little amount of LAA. The four-point resistance measurement shows that the resistivity of the

aerogels decreases dramatically with increasing LAA concentration (Supplementary Table 1).”

6. In Supplementary Fig. 2a and b, the y-coordinate should be shortened.

Response: Thank you and the relevant figures are improved accordingly in the revised supplementary information and the Supplementary Fig. 2 has been updated to the Supplementary Fig. 4 (Page 10, line 178).

Supplementary Fig. 4 a, I-V characteristics of the osmotic power sources without RTIL (Au/GO/rGO/Au) and with RTIL (Au/GO/ RTIL/rGO/Au) at 25°C. **b**, Cyclic voltammogram (CV) of the osmotic power sources without RTIL (Au/GO/rGO/Au) and with RTIL (Au/GO/ RTIL/rGO/Au) at the scan rate of 0.1 mV/s at 25°C.

7. In the figure, the spaces are required between the number and the unit, such as in Fig. 1b, 1c, Fig. 3b and 3c.

Response: Thank you and the relevant figures are improved accordingly in the revised manuscript. Detailed revision is as below. In the revised manuscript:

Fig. 1 b, SEM image of the GO layer (cross section). **c**, SEM image of the rGO layer (cross section).

Fig. 3 b, SEM image of the GO aerogel. Inset photo of the obtained GO aerogel. **c**, SEM image of the GO+KOH aerogel. Inset is the EDS element mapping of the GO+KOH aerogel.

8. In Fig. 3b and 3c, the SEM images of the GO layer and rGO (cross section) should be a unified scale for the observation of graphene layered structure. The scale bar in Supplementary Fig. 3 a and b should be added.

Response: Thank you and the relevant figures are improved accordingly in the revised manuscript and supplementary information, and the Supplementary Fig. 3 has been updated to the Supplementary Fig. 5. Detailed revision is as below.

In the revised manuscript:

Fig. 3 b, SEM image of the GO aerogel. Inset photo of the obtained GO aerogel. **c**, SEM image of the GO+KOH aerogel. Inset is the EDS element mapping of the GO+KOH aerogel.

In the revised supplementary information (Page 12, line: 214):

Supplementary Fig. 5 a-b, The optical image of the AgNO₃/GO boundary before (a) and after (b) the power source was discharged.

9. In Supplementary Figure 2g, the EIS data of the osmotic power sources without RTIL is confused, and we can't get the conclusion about the charge transfer resistance.

Response: Thanks for the reviewer's precious suggestion and the Supplementary Fig. 2 has been updated to the Supplementary Fig. 4. Equivalent circuit of the osmotic power source with and without RTIL has been fitted for Supplementary Fig. 4h in the revised supplementary information as follow. Details were explained in answering Q11.

In the revised supplementary information (Page 10, line 178):

Supplementary Fig. 4h, Electrochemical impedance spectrum (EIS) of the osmotic power sources with RTIL (Au/AgNO₃/GO/RTIL/rGO/Au) and without RTIL (Au/AgNO₃/GO/rGO/Au). The (CR)W equivalent circuit (Inset schematic) was used to fit the power source, where R is the charge transfer resistance, C is the double-layer capacitance, and W is the Warburg impedance. The fitted charge transfer resistance of the device without and with RTIL is $1.887 \times 10^5 \Omega$ and $2.679 \times 10^4 \Omega$ respectively.

10. The authors mentioned, the structure of GO aerogel is nanopores with diameter of several nanometers, but in Figure 3b, there are pores with diameter of hundreds of micrometers.

Response: Indeed, Fig.3b is the SEM image of the as-made GO aerogel and the inset in Fig.3b shows the morphology of the porous GO aerogel. When the GO aerogel was used in the osmotic power device, it was pressed and the diameter of several nanometers refers to the pressed state of GO aerogels in the device. In addition, the thickness of the aerogels also influences the output performance of the power source. The optimized thickness of the aerogel was found to be 15 μm as reported in the revised manuscript. The corresponding discussion has been added in the revised manuscript. Detailed revision is as below.

In the revised manuscript (Page 13, line 268-270):

“It should be noticed that inset in Fig.3b is the image of the as-made GO aerogel. When the GO aerogel was used in the osmotic power device, it was pressed and the thickness of the aerogels also influences the output performance of the power source.”

11. For the EIS data, the equivalent circuit diagram should be fitted first, and then to compare the impedance value. So, the EIS part in the manuscript should be revised.

Response: Thanks for the reviewer's precious suggestion, as shown in Fig. R1, the equivalent circuit has been fitted based on the EIS data in Supplementary Fig. 4h. The (CR)W equivalent circuit (Inset schematic in Fig. R1) was used to fit the power source, where R is the charge transfer resistance, C is the double-layer capacitance, and W is the Warburg impedance.

Fig. R1 Comparison of the fitted curve with the original EIS data for the osmotic power source without RTIL and with RTIL respectively. Inset is the equivalent circuit diagram of (CR)W.

Table R1 Comparison of the fitted results based on the osmotic power source with and without RTIL.

Device state	Charge transfer resistance (Ω)
without RTIL	1.887×10^5
with RTIL	2.679×10^4

Table R1 listed the fitted charge transfer resistances of the device without and with RTIL, which is $1.887 \times 10^5 \Omega$ and $2.679 \times 10^4 \Omega$ respectively. The fitted curves were compared with the original curves in Fig. R1, where the lines are the fitted results. The corresponding discussion has been added in the revised Supplementary Fig. 4h and the

figure caption.

The corresponding discussion has been added in the revised manuscript. Detailed revision is as below.

In the revised manuscript (Page 8, line 155-157):

“Furthermore, electrochemical impedance spectrum (EIS) confirmed the charge transfer resistance decreased significantly with addition of RTIL as shown in Supplementary Fig. 4h.”

12. Whether the Voc of the device can be further increased by changing the redox potential by other electrochemical reaction?

Response: Yes, indeed, the Voc of the device can be tuned by changing the redox potential by other electrochemical reactions. We carried on further experiments, replacing AgNO₃ with a redox couple of potassium ferricyanide. The Voc of the power source is composed of the voltage from ion gradient diffusion (E_{diff}) and the redox reactions (E_{redox}) at the Au charge collector interface.

$$V_{OC} = E_{diff} + E_{redox}$$

As discussed in the Supplementary material, E_{diff} is 1.2 V.

The potential could be estimated with the guide of the Nernst equation.

$$E_{redox} = E^0_c - E^0_a = 0.16 \text{ V}$$

$$V_{OC} = E_{diff} + E_{redox} = 1.2 + 0.16 = 1.36 \text{ V}$$

The theoretical value Voc of 1.36 V is in good accordance of the experimental data of 1.3 V in Fig. R2. Compared with the AgNO₃ redox reaction (1.5 V as shown in Supplementary Fig. 4e), Fig. R2 showed that the Voc can be tuned to 1.3 V by introducing the potassium ferricyanide redox reaction. It also means that such osmotic power source could provide a paradigm where interfacial electrochemical reactions could be tuned to tailor the output of the RED cell.

Fig. R2 I-V characteristics of the planar osmotic power source with potassium ferricyanide redox reactions at the interface between GO and Au charge collector.

The corresponding discussion has been added in the revised manuscript. Detailed revision is as below.

In the revised manuscript (Page 7-8, line 154-155):

“It also provides a paradigm where the Voc of the osmotic power source could be tuned by tailoring interfacial electrochemical redox reactions.”

13. Many recent publications related to nanofluidic iontronics have been published, such as *Science*, 2021, 373: 628-629, *Chinese Chemical Letters*, 2021, 32: 642-648, *Nano Today*, 2020, 33: 100868, *Nano Letters*, 2020, 20, 6937-6946, *Advanced Materials*, 2019, 31: 1805130, et al. In order to help readers better understand the important of this work, these references should be cited, correspondingly.

Response: Thank you and all these references were added accordingly in the revised manuscript. Detailed revision is as below.

In the revised manuscript (Page 2, line 23-27):

“Iontronics couple the electron/ion charge transfers and exchange signals at the interface of electronic/ionic conductors, differentiating them from most electronics using just electrons and/or holes as the dominating charge carriers³⁻⁷. Bioinspired nanofluidic iontronics could have compatible signals with neurons to enable implantable iontronic devices or even neuronal-computer interfaces³.”

(Reference updates):

3. Hou, Y. & Hou, X. Bioinspired nanofluidic iontronics. *Science* 373, 628-629 (2021).

4. Zhang, S., Boussouar, I. & Li, H. Selective sensing and transport in bionic nanochannel based on macrocyclic host-guest chemistry. *Chin. Chem. Lett.* 32, 642-648 (2021).

5. Zhan, K. et al. Tannic acid modified single nanopore with multivalent metal ions recognition and ultra-trace level detection. *Nano Today* 33, (2020).

6. Wang, M., Hou, Y., Yu, L. & Hou, X. Anomalies of Ionic/Molecular Transport in Nano and Sub-Nano Confinement. *Nano Lett.* 20, 6937-6946 (2020).

7. Wang, M. et al. Dynamic Curvature Nanochannel-Based Membrane with Anomalous Ionic Transport Behaviors and Reversible Rectification Switch. *Adv. Mater.* 31, e1805130 (2019).

I will be happy to recommend for publication a revised version of the manuscript in *Nature Communications*.

Response: We sincerely thank the reviewer for appreciating and recommending our work for publication in *Nature Communications*. We have revised the manuscript according to the reviewer's precious suggestions point by point.

Finally, we would like to thank the reviewer again for all these valuable comments and for the thoughtful and careful review towards improving our manuscript.

Reviewer #2 (Remarks to the Author):

This is an interesting paper related with flexible iontronics based on nanofluidic materials. They developed novel foldable power source by using iontronics. The work could be publishable in Nature Communications with the following minor correction.

Response: We are grateful to the reviewer for the thorough review and help to improve the paper. We are appreciated the comments from the Reviewer and the point-to-point responses are listed in the following:

1) Lego-like osmotic power source: The reviewer thinks the word "Stacked" is better than the word "Lego-like".

Response: We thank the reviewer for pointing out a more appropriate way to describe the power source. In the revised manuscript after carefully consideration, we have used “modular design” to replace the “*Lego-like*” that was broadly used in the revised manuscript. The power source using the self-healing ionogels not only involves ‘stacking’ but also ‘separation’, which could plug and unplug to change the counter electrode with different concentration of ions. Thus “modular design” may be a more appropriate term than our original description of “*Lego-like*”. In the revised manuscript (Line 338, 347, 375, 405, 441, 616), the “Lego-like” had been replaced by “modular design”.

2) Stability of the device needs to be mentioned.

Response: Firstly, all batteries started discharge as soon as they were assembled, and the shelf-life for commercial lithium ion batteries is about 5 years. This is particularly true for the concentration-gradient based osmotic power source. However, this is also the unique benefit of the osmotic power source in that if it is packed separately or stored in vacuum or in a condition without any humidity, it will keep its energy and extend its

shelf-life as long as we wish. Experiments has been carried out and our osmotic power source could not generate any power (V_{oc} is zero) in the glovebox where the content of water could be ignored ($H_2O < 0.5$ ppm). It was kept in the glovebox for half year, but when it was taken out, the V_{oc} is still around 1.4 V (supplementary Fig. 5g), which almost reached the V_{oc} (1.5 V) as the freshly made osmotic power source (Supplementary Fig. 4d). Its power density kept value of 26 mW cm^{-3} (Supplementary Fig. 5h) comparable to the freshly made osmotic power source (28 mW cm^{-3} in Fig. 1e). Secondly, the stability of the conventional osmotic power source relied on the humidity, but in our case, application of RTIL accelerates the kinetics of ion diffusion in nanoconfined channels and it also imparts the osmotic power source to overcome the operation limitation on humidity and temperature. The stability of the device is much more improved than the ones depending on purely humidity. In addition, the RTIL used in the device is very stable at ambient environment with a broad electrochemical window. Unlike organic or aqueous electrolyte in batteries, RTIL will not evaporate away from the power source.

The corresponding discussion has been added in the revised manuscript. Detailed revision is as below.

In the revised manuscript (Page 9, line 182-194):

“All batteries started discharge as soon as they were assembled, and this is particularly true for the concentration-gradient based osmotic power source. However, it is also the unique benefit of the osmotic power source in that if it is packed separately or stored in vacuum or in a condition without any humidity, it will keep its energy and extend its shelf-life as long as we wish. Experiments has been carried out and our osmotic power source could not generate any power (V_{oc} is zero) in the glovebox (Supplementary Fig. 5f) where the content of water could be ignored ($H_2O < 0.5$ ppm). It had been kept in the glovebox for half year, but when it was taken out, the V_{oc} is still around 1.4 V (Supplementary Fig. 5g), which almost reached the V_{oc} (1.5 V) as the freshly made osmotic power source (Supplementary Fig. 4d). Its power density kept value of 26 mW cm^{-3} (Supplementary Fig. 5h) comparable to the freshly made osmotic power source (28 mW cm^{-3} in Fig. 1e). In addition, besides accelerating the kinetics of ion diffusion

in nanoconfined channels, the RTIL itself used in the device is very stable at ambient environment with a broad electrochemical window. Unlike organic or aqueous electrolyte in batteries, RTIL will not evaporate away from the device.”

Supplementary Fig. 5 f, Voc of the device was zero in the glovebox ($H_2O < 0.5$ ppm). **g**, Voc of the power source that had been kept in glovebox for half year was measured at ambient environment. **h**, The I-V characteristics and power density of the power source that was kept in glovebox for half year.

3) Conductivity of rGO needs to be provided.

Response: As the reviewer suggested, we carried out the electrical conductivity test for rGO. The electrical conductivity was measured on the deposit rGO film across two Au electrodes. The size of rGO film was $2 \text{ mm} \times 5 \text{ mm} \times 0.01 \text{ mm}$ and its mean resistance (R) was measured to be $0.7 \text{ M}\Omega$ (with 3 measurements). Thus, the electrical conductivity of rGO was calculated to be about 0.06 S m^{-1} using the formula of $\sigma = L/(R \times A)$, where L and A represent the thickness and the area of the rGO film. Unlike the sharp increase in conductivity by the chemical reduction from hydrazine or vitamin C ($2690 - 9960 \text{ S m}^{-1}$), the KOH is not an efficient reductant for GO ($0.02\text{-}1.55 \text{ S m}^{-1}$)⁴, but could partially remove the oxygen-containing groups of GO sheets through a series of deoxygenation reactions^{2,3}. More importantly, rGO from KOH is an effective cation (K^+) reservoir to form the ion gradient with GO, which is the essential part of this paper.

The corresponding discussion has been added in the revised manuscript. Detailed revision is as below.

In the revised manuscript (Page 4, line 77-78):

“The electrical conductivity of rGO was measured to be about 0.06 S m^{-1} , falling in the conductivity range of rGO by KOH treatment ($0.02\text{-}1.55 \text{ S m}^{-1}$)^{38,39}.”

4) It is better to cite the following references for graphene iontronic pressure sensor.

- Kim et al., *Adv. Funct. Mater.* 30, 2070089(2020)

- Nie et al., *Adv. Mater.* 27, 6055 (2015)

Response: Thank you and all these references were added accordingly in the revised manuscript. Detailed revision is as below.

In the revised manuscript (Page 2, line 27–28; Page 5, line 108-110):

“Enhanced sensitivity of tactile sensor⁸ and pressure sensor⁹ could also be obtained by iontronic films.”

“Although the voltage of such osmotic power source was high, the current was low as it purely came from the ion gradient and there was no Faradaic reaction when using Au as charge collectors, however, addition of RTILs was found to boost up the current^{8,9,40}.”

(Reference updates):

8. Kim, J. S. et al. Iontronic Graphene Tactile Sensors: Enhanced Sensitivity of Iontronic Graphene Tactile Sensors Facilitated by Spreading of Ionic Liquid Pinned on Graphene Grid. *Adv. Funct. Mater.* 30, (2020).

9. Nie, B. et al. Flexible transparent iontronic film for interfacial capacitive pressure sensing. *Adv. Mater.* 27, 6055-62 (2015).

Finally, we would like to thank the reviewer again for these valuable comments and for the thoughtful and careful review towards improving our manuscript.

Reviewer #3 (Remarks to the Author):

This manuscript of “Flexible iontronics based on 2D nanofluidic material” reports on a GO-based iontronic device that can harvest energy with a high power density compared to those have already been published. The performance of energy conversion can be boosted by incorporating: asymmetric distribution of surface charges of GO/rGO, 3D gel structure, and the addition of RTILs. I like the Lego-like design that consists of GO aerogels and GO+KOH layers sandwiching a self-healing gel. The ultrathin and wearable features make the real application possible. The device can even work in harsh environments. This manuscript is informative and should be of interest to a broad audience. I'm willing to support its publication after the following comments have been addressed.

Response: We are grateful to the reviewer for the thorough review and help to improve the paper. The point-to-point responses are listed in the following:

1. The authors adopt previous work to claim 2 nm is optimal for energy conversion. However, continuum-based theory breaks down for confinement less than ~ 3 nm and may become invalid for the system with RTILs. This explanation may be OK for this paper but a brief justification is necessary.

Response: Thank you for the comments and the detail justification is addressed. Unusual behavior of ion transport kinetics in channels narrower than the Debye length of electrolyte has been observed, the surface charges on the inner walls of nanofluidic channels repel ions of the same charge and attract counter ions, making them the dominating charge carriers. Such unipolar ion transport can enhance ionic conductivity up to several orders of magnitude, which breaks the conventional continuum-based theory. Osmotic energy conversion could be boosted by such 3D aerogel interface, when pore diameter of around 2 nm or less is in the order of Debye screening length in the 2D nanofluidic channels. RTIL enhances charging dynamics under such severe

nanoconfinement⁵ and accelerates ionic conductivity in the nanopores.

The corresponding discussions have been added in the revised manuscript. Detailed revision is as below.

In the revised manuscript (Page 2, line 28-32; Page 12, line 250-258; Page 14, line 293-295):

“Unusual behavior of ion transport kinetics in channels narrower than the Debye length of electrolyte has been observed, the surface charges on the inner walls of nanofluidic channels repel ions of the same charge and attract counter ions, making them the dominating charge carriers¹⁰. Such unipolar ion transport can enhance ionic conductivity up to several orders of magnitude, which breaks the conventional continuum-based theory¹¹.”

“GO as 2D nanofluidic material with negative surface charge has the enhanced diffusion related to the strong ion-ion correlations under severe nanoconfinement. In this case, charge is transported through the pore by the flux of cations that interact with the immobile negative surface charges from carboxyl groups etc. Osmotic energy conversion could be boosted by such 3D aerogel interface, when pore diameter of around 2 nm or less is in the order of Debye screening length in the 2D nanofluidic channels. The Poisson-Nernst-Planck (PNP) model⁵⁰ was used to simulate the osmotic potential from the ion gradient of the 2D nanofluidic material and it also showed that the appropriate pore size for membrane-scale osmotic energy conversion should be around 2 nm, matching our experimental results.”

“RTIL enhances charging dynamics under severe nanoconfinement and accelerates ionic conductivity in the nanopores avoiding the humidity limitation for the osmotic power source and could even further overcome temperature limitations.”

2. The authors propose surface charge is essential for osmotic energy. While Fig. S1a shows a surface-charge-governed feature, it is difficult to determine the charge density of porous membranes through ionic conductance measurement. The plateau may result from other factor such as contaminations. Did authors measure the zeta potential of the GO membrane to confirm its charge property? Besides, Fig. S1 was

performed with KCl solution rather than the solution used in the main text. Did authors observe the similar nonlinear conductance under the experimental condition, for example, KOH with RTIL?

Response: Thank for the reviewer's suggestion and the Supplementary Fig. 1 has been updated to the Supplementary Fig. 2. The zeta potential was measured (Supplementary Fig. 2a), and reveals GO was negatively charged due to the abundant oxygen-containing functional groups, such as carboxyl and hydroxyl etc. When forming the GO film from GO solution, only restacking of GO sheets happens and there are no chemical reactions involved and these oxygen-containing functional groups would not be removed. Thus, the surface of the GO flake should also be negatively charged.

The plateau in Supplementary Fig. 2b indicated the K^+ could transport through the 2D nanofluidic channels of the deposited GO film. As the reviewer suggested, we carried out the experiment and observed the similar nonlinear conductance under the experimental condition with the $CaCl_2$ solution (Supplementary Fig. 2c). Such nonlinear conductance matched the previous reports from J. Huang et al. ⁶, and was verified to be originated from the cation transport through the 2D nanofluidic channels instead of from other contaminations. We could not observe nonlinear conductance feature with KOH and its conductance changes dynamically during measurement, because KOH will partially remove the oxygen-containing groups of GO sheets through a series of deoxygenation reactions^{2,3}. So, when KOH solution contacts the GO, deoxidizing reaction occurs and it will break the 2D nanofluidic channels. Indeed, the KOH was mainly used to form rGO in this paper, which was an effective cation (K^+) reservoir to form the ion gradient with GO.

The corresponding discussion has been added in the revised manuscript. Detailed revision is as below.

In the revised manuscript (Page 4-5, line 86-95):

“As shown in Supplementary Fig. 2a, the zeta potential of the GO solution reveals GO was negatively charged due to the abundant oxygen-containing functional groups. When forming the GO film from GO solution, only restacking of GO sheets happens

and there are no chemical reactions involved and these oxygen-containing functional groups would not be removed. Thus, the surface of the GO flake should also be negatively charged. Ionic conductivity as a function of salt concentration was measured and the plateau in Supplementary Fig. 2b indicated the K^+ could transport through the 2D nanofluidic channels inside the deposited GO film. Similar conductivity curve was also observed under the experimental condition with the $CaCl_2$ solution (Supplementary Fig. 2c). Such nonlinear conductance matched the previous reports from J. Huang et al.¹¹, and was verified to be originated from the cation transport through the 2D nanofluidic channels inside GO instead of from contaminations.”

Supplementary Fig. 2a, Zeta potential of GO aqueous solution (5mg/mL, pH 4). Error bar represents standard deviations for 3 measurements. **b**, Ionic conductivity as a function of KCl concentration measured through 2D nanofluidic channels of GO. The conductivity of bulk solution is shown as benchmark. **c**, Ionic conductivity as the function of $CaCl_2$ concentration measured through 2D nanofluidic channels of GO. The conductivity of bulk solution is shown as benchmark.

3. How do the authors obtain the ion selectivity of 0.99 in Supplementary Discussion? It seems the values for activities are not provided. It is important to clarify the calculation of activity. Ionic activity in ionic liquids could be complicated because of the significant short-range correlation. In that case Debye-Huckel theory fails to describe enhancement of activity at high concentrations.

Response: Thank you for the very insightful points. It was reported that graphene nanopores⁷ were found to preferentially transporting K^+ over its counter anions such as

Cl⁻ with selectivity ratios over 100 and hydrated K⁺ diffuses orders magnitude more quickly than most hydrated ions within the 2D nanofluidic channels. Based on such prior art⁷, we calculated the ion selectivity of GO to K⁺ is 0.99. Indeed, the Debye-Huckel theory applies best in the diluted solution of the electrolyte. However, RTIL was drop casted on the junction between GO and rGO, which mainly accelerates charging dynamics in 2D nanofluidic channels of GO⁵. The osmotic power source is operated as a solid state power source in contrast to those in traditional electrolyte. The measured E_{diff} is quite high, also indicating a very high selectivity of K⁺ cations.

The Supplementary Discussion had been revised as follow. In the revised supplementary information (Page 3, line: 52-59):

“It was reported that graphene nanopores¹ were found to preferentially transporting K⁺ over its counter anions such as Cl⁻ with selectivity ratios over 100 and hydrated K⁺ diffuses orders magnitude more quickly than most hydrated ions within the 2D nanofluidic channels. Based on such prior art¹, we calculated the ion selectivity of GO to K⁺ is 0.99. The charge selectivity of GO to K⁺ is very high (close to 0.99), and the E_{diff} for K⁺ in the osmotic power sources (both the planar one and the 3D GO aerogel one) follow the same equation. The osmotic power source is operated as a solid state power source in contrast to those in traditional electrolyte. The measured E_{diff} is quite high, also indicating a very high selectivity of K⁺ cations.”

4. (following above) 0.99 means a nearly perfect ion selectivity. It's puzzling because the surface charge density of GO membrane was reported as $\sim 2 \text{ mC/m}^2$ (Zhang et al., Nat Commun 10, 1253 (2019)). This indicates that the electrostatic repulsion is not such strong. Is there other mechanism, for example, steric exclusion or interactions between RTIL and GO surface?

Response: GO membranes containing 2D nanofluidic structures possess both horizontal and vertical channels with negative surface charge for fast cation transport. The surface charge density of GO membrane reported in Zhang's paper⁸ was measured when ions transport in the vertical direction, in contrast, the K⁺ ions transport in the

horizontal direction within 2D nanofluidic channels in our manuscript (Fig. R3).

It was reported that ions transport much faster in the horizontal direction than in the vertical direction in GO membranes (Fig. R3) and planar confinement in such 2D nanofluidic materials expands translational degrees of freedom for ionic transport engendering unusual ion dynamics^{9,10} As pointed out by the reviewer, the faster transport of ions in the horizontal direction may be contributed by steric exclusion and charge transport acceleration from RTIL besides simply electrostatic repulsion.

Fig. R3 Ion transport direction through the GO membrane by vertical transport reported in Zhang’s paper⁸ and horizontal transport reported in our manuscript.

The corresponding discussion has been added in the revised manuscript. Detailed revision is as below.

In the revised manuscript (Page 3-4, line 65-68):

“Planar confinement in 2D nanofluidic material of GO expands translational degrees of freedom for ionic transport engendering unusual ion dynamics and ions transport much faster in the horizontal direction within 2D nanofluidic channels than in the vertical direction in the GO film^{14,15}.”

Zhang’s paper has also been cited in the revised manuscript. Detailed revision is as below. In the revised manuscript (Page 2, line 36-37):

“Graphene oxide (GO) as 2D nanofluidic material with negative surface charge from carboxyl and hydroxyl groups etc. has shown special affinity to water¹⁷ and controllable

ion transport properties¹⁸.”

(Reference updates):

18. Zhang, M. et al. Controllable ion transport by surface-charged graphene oxide membrane. *Nat. Commun.* 10, 1253 (2019).

5. Line 296, a power density of 1.3 mW cm⁻² was obtained from Lego-like power source and was compared to commercialization benchmark of 0.5 mW cm⁻². Is the tested surface area 0.6 cm × 0.2 cm, according to Supporting Information? If so, this number is still below realistic conditions of commercial ion-exchange membranes. Since the power density is highly dependent on the membrane area (Xiao et al., *Joule* 3, 2364–2380, 2019), the tested area should be specified when making a comparison.

Response: Area of 0.60 cm × 0.20 cm refers to the surface area of the planar device, but not the ‘Lego-like power source’. The ‘Lego-like power source’ was made from the RTIL ionogel, and we can make it in a more compact space with area of 0.32 cm × 0.20 cm. The power density of it was shown in Fig.4b and the maximum power of 84 μW can be calculated and the areal power density is about 1.3 mW cm⁻². Indeed, as the reviewer pointed out the power density is highly dependent on the membrane area and 3D porous membranes could be used to construct higher output power source. We tried to scale up the 3D device by increasing salinity gradient and increasing the area as shown in Fig. 4e. The stacked osmotic power source could light up electrochromic devices and liquid crystal display screens.

We thank the reviewer’s valuable comments and the tested area has been specified in the section of Supplementary Discussion. Detailed revision is as below. In the revised Supplementary information (Page 4, line: 72-81):

“Calculation of areal power density of the 3D osmotic power source

The energy density of the osmotic cell can be calculated by

$$E = I \int_0^t U$$

where I , U , t are the discharge current, the electric potential and discharge time, respectively.

The areal energy density (E_s) can be expressed as

$$E_s = E/S$$

where S represent the surface area of the GO device.

The 3D osmotic source was made from the RTIL ionogel, which enables it made in a more compact space with area of $0.32 \text{ cm} \times 0.20 \text{ cm}$. The maximum power of $84 \mu\text{W}$ can be calculated from Fig. 4b and the areal power density is about 1.3 mW cm^{-2} .”

Xiao’s paper has also been cited in the revised manuscript. Detailed revision is as below.

In the revised manuscript (Page 1-2, line 21-23; Page 18, line 375-377):

“The osmotic energy could be generated based on either pressure-retarded osmosis (PRO) or reverse electrodialysis (RED)¹, and the ion regulation component is the critical part for such power generation².”

“Such modular design power source offers the areal power density of 1.3 mW cm^{-2} at ambient environment that is significantly higher than 0.5 mW cm^{-2} , which has been flagged as the target for making salinity gradient power economically viable².”

(Reference updates):

2. Xiao, K., Jiang, L. & Antonietti, M. Ion Transport in Nanofluidic Devices for Energy Harvesting. *Joule* 3, 2364-2380 (2019).

6. Self-healing materials have been used to make advanced membranes in water filtration. Fig. S6 demonstrates the self-healing property for ionogels but I’m curious whether the authors compare the output power before and after the cut. Would the power generation be affected a lot after healing? Also, it would be better if typical self-healing test (i.e., stress-strain curve) can be provided. Longer-term performance of self-healed power generator would significantly elevate this work.

Response: We thank for the reviewer’s suggestion. Firstly, we carried out the

impedance measurement for the self-healing ionogel before and after being cut (Fig. R4a). Then, the ionic conductivity of the ionogel was evaluated by EIS and indicated that there was no obvious difference for the ionic conductivity of ionogel before and after being cut (Fig. R4b), which remained in the same orders of magnitude (10^{-3} S cm^{-1}). This might be attributed to the highly reversible ion-dipole interactions of the ionogel¹¹.

Secondly, the output power of the modular design osmotic power source was also measured and compared for the self-healing ionogel before and after being cut. As shown in Fig. R4c, the modular design power source with the ionogel after healing has the V_{OC} of 0.62 V and the I_{SC} of 410 μA . Compared with the volumetric specific power density 12 mW cm^{-3} of the power source with ionogel before being cut (Fig. 4b), its power density is still around 11.8 mW cm^{-3} . So, the power generation of the modular design power source is not affected a lot after healing.

Fig. R4 The electrical characterization of self-healing ionogels. **a**, The complex plane plot of the ionogels before and after being cut. **b**, The ionic conductivity of the ionogels before and after being cut. Error bar represents standard deviations for 3

measurements. **c**, I-V characteristics and power density of the modular design power source with ionogel after healing. **d**, I-V characteristics and power density of the modular design power source with aerogel and ionogel components stored half year in the glovebox.

For the long-term performance, as mentioned in the manuscript, the component parts of the modular design 3D osmotic power source could be stored separately before use, avoiding self-discharge. We carried out the further experiments using the component parts such as the cathode GO aerogels, ionogels and anode rGO aerogels, which had been kept separately in the glovebox for half year. The cathode GO aerogel electrode and anode rGO aerogels were adhered together by ionogels through the self-healing process. Compared with the power source with freshly-made materials (Fig. 4b), similar V_{oc} and I_{sc} is obtained of 0.6 V and 400 μA respectively and the volumetric specific power density is also around 11.1 $mW\ cm^{-3}$ (Fig. R4d).

Fig. R5 Mechanical self-healing abilities. **a**, Stress-strain curves of original and healed ionogels. **b**, Young's modulus of original and healed ionogels. Error bar represents standard deviations for 3 measurements.

As suggested by the reviewer, we also carried out the mechanical test using an Instron E3000 instrument. The tensile experiments were performed at 25°C at a strain rate of 5 $mm\ min^{-1}$. Each mechanical test was repeated with 3 individual samples. Considering the restoration of both stress and the strain, the mechanical self-healing efficiency was defined as the proportion of toughness restored relative to the original toughness¹².

Compared with the pristine ionogels before being cut, the healed ionogels maintained the similar Young's modulus and tensile stress curves (Fig. R5). Our self-healing ionogels shows characteristic feature of the stress-strain curve of self-healing ionogels reported by other groups^{11,13}.

The corresponding discussion has been added in the revised manuscript. Detailed revision is as below.

In the revised manuscript (Page 15-16, line 326-337; Page 16-17, line 349-359):

“Such ionogel could be cut and then placed together, exhibiting fast self-healing capability at ambient environment shown in Supplementary Fig. 10a. The tensile experiments were performed by an Instron E3000 instrument at 25°C at a strain rate of 5 mm min⁻¹. Considering the restoration of both stress and strain, the mechanical self-healing efficiency was defined as the proportion of toughness restored relative to the original toughness⁵⁵. As shown in Supplementary Fig. 10b, the self-healing ionogel shows characteristic feature of the stress-strain curve of self-healing ionogels reported by other groups^{52,53}. Compared with the pristine ionogels before being cut, the healed ionogels maintained similar Young's modulus and tensile stress curves (Supplementary Fig. 10b-c). The ionic conductivity of the ionogel was evaluated by EIS (Supplementary Fig. 10d) and indicated that there was also no obvious difference for the ionic conductivity of ionogel before and after being cut (Supplementary Fig. 10e), which remained in the same orders of magnitude (10⁻³ S cm⁻¹). Such effective self-healing capability might be attributed to the highly reversible ion-dipole interactions of the ionogel⁵³.”

Supplementary Fig. 10 Mechanical and electrical properties of the self-healing ionogel. **a**, Photo of the self-healing demonstration of an ionogel at room temperature. **b**, Stress-strain curves of the original and healed ionogels. **c**, Young's modulus of the original and healed ionogels. Error bar represents standard deviations for 3 measurements. **d**, The complex plane plot of the ionogels before and after being cut. **e**, The ionic conductivity of the ionogels before and after being cut. Error bar represents standard deviations for 3 measurements.

“Proof of concept of the plug-unplug modular design in the osmotic power source was demonstrated by comparing the output power for the self-healing ionogel before and after being cut. As shown in Supplementary Fig. 11a, the modular design power source with the ionogel after healing has the V_{oc} of 0.62 V and the I_{sc} of 410 μ A with its power density still around 11.8 $mW\ cm^{-3}$ vs. 12.0 $mW\ cm^{-3}$ (Fig. 4b) from the pristine device. Thus the power generation of the modular design power source is not affected a lot after healing. For the long-term performance, further experiments were carried out using the component parts such as the cathode GO aerogels, ionogels and anode rGO aerogels,

which had been kept separately in the glovebox for half year. The cathode GO aerogel electrode and anode rGO aerogels were adhered together by ionogels through the self-healing process. Similar V_{oc} and I_{sc} are obtained and the volumetric specific power density is around 11.1 mW cm^{-3} (Supplementary Fig. 11b), showing perfect long-term performance.”

Supplementary Fig. 11 I-V characteristics and power density of the modular design power source **a**, with ionogel after healing. **b**, with aerogel and ionogel components stored half year in the glovebox.

7. Here are several minors:

-Improve the resolution of Fig 1f

-Line 46, it states that power densities of RED are small, but the cited ref 21,26 are harvesting energy through RED. Generally, “surface” power density (e.g., W/m^2) is commonly used in RED power generation.

Response: Thank you and the relevant figure has been improved accordingly in the revised manuscript. Detailed revision is as below.

In the revised manuscript:

Fig. 1 f, Comparison of energy and power density of Au/AgNO₃/GO/RTIL/rGO/Au in the Ragone plot (Data for lithium thin-film batteries, supercapacitors and Al electrolyte capacitors are collected from references^{47,48}).

As the reviewer suggested, areal power density (e.g., W m²) is commonly used in RED power generation, but to compare with other power sources such as batteries and supercapacitors in Ragone plot of Fig. 1f, volumetric power and energy densities were calculated. The volumetric power densities from the original cited ref. 21,26 (ref. 30,35 in the revised manuscript) were either reported from their original paper or calculated with the given information in the paper. To make the comparison of the osmotic power source output in areal power density (mW cm⁻²), the outputs of various osmotic power sources with various materials were listed in the Supplementary Table 2 and added in the revised manuscript. Detailed revision is as below.

In the revised manuscript (Page 18, line 377-379):

“The outputs (in mW cm⁻²) of various osmotic power sources with different materials were compared in Supplementary Table 2, and our modular design power source generates by far the highest areal power density among them.”

Supplementary Table 2 Comparison of the modular design osmotic power source output with various materials.

Type	Materials	Power density (mW cm ⁻²)	Reference
This work	GO	1.3	
2D materials	GO	0.526	Ref. 2
	GO	0.077	Ref. 3
	GO	0.035	Ref. 4
	rGO	0.115	Ref. 5
	Mxene	0.21	Ref. 6
	Mxene/Kevlar	0.41	Ref. 7
Polymer	PES-Py/PAEK-HS	0.266	Ref. 8
	PMA/PS-b-P4VP	0.38	Ref. 9
	Polyimide	0.026	Ref. 10
Others	Silicon	0.77	Ref. 11
	SNF/AAO	0.286	Ref. 12
	Hydrogel/ANF	0.506	Ref. 13
	Mesoporous carbon/AAO	0.346	Ref. 14

Finally, we would like to thank the reviewer again for these valuable comments and for the thoughtful and careful review towards improving our manuscript.

References

1. Cao, L. et al. On the origin of ion selectivity in ultrathin nanopores: Insights for membrane-scale osmotic energy conversion. *Adv. Funct. Mater.* **28**, 1804189 (2018).
2. Fan, X. et al. Deoxygenation of Exfoliated Graphite Oxide under Alkaline Conditions: A Green Route to Graphene Preparation. *Adv. Mater.* **20**, 4490-4493 (2008).
3. Pei, S. & Cheng, H.-M. The reduction of graphene oxide. *Carbon* **50**, 3210-3228 (2012).
4. Fernández-Merino, M. J. et al. Vitamin C Is an Ideal Substitute for Hydrazine in the Reduction of Graphene Oxide Suspensions. *J. Phys. Chem. C* **114**, 6426-6432 (2010).
5. Kondrat, S., Wu, P., Qiao, R. & Kornyshev, A. A. Accelerating charging dynamics in subnanometre pores. *Nat. Mater.* **13**, 387-93 (2014).
6. Raidongia, K. & Huang, J. Nanofluidic ion transport through reconstructed layered materials. *J. Am. Chem. Soc.* **134**, 16528-31 (2012).
7. Rollings, R. C., Kuan, A. T. & Golovchenko, J. A. Ion selectivity of graphene nanopores. *Nat. Commun.* **7**, 11408 (2016).
8. Zhang, M. et al. Controllable ion transport by surface-charged graphene oxide membrane. *Nat. Commun.* **10**, 1253 (2019).
9. Chmiola, J. et al. Anomalous increase in carbon capacitance at pore sizes less than 1 nanometer. *Science* **313**, 1760-3 (2006).
10. Futamura, R. et al. Partial breaking of the Coulombic ordering of ionic liquids confined in carbon nanopores. *Nat. Mater.* **16**, 1225-1232 (2017).
11. Cao, Y. et al. Self-healing electronic skins for aquatic environments. *Nature Electronics* **2**, 75-82 (2019).
12. Tee, B. C. K., Wang, C., Allen, R. & Bao, Z. An electrically and mechanically self-healing composite with pressure- and flexion-sensitive properties for electronic skin applications. *Nat. Nanotechnol.* **7**, 825-832 (2012).

13. Cao, Y. et al. A transparent, self-healing, highly stretchable ionic conductor. *Adv. Mater.* 29, (2017).

REVIEWERS' COMMENTS

Reviewer #1 (Remarks to the Author):

In this revised manuscript, the authors still need to consider the following comments.

1. For the EIS data in Supplementary Fig. 4h, it is too confused to fit the equivalent circuit. At the same time, the test conditions should be stated in the text so that we can analyze the state inside under this condition.

2. For our previous comment 10, the authors said "the diameter of several nanometers refers to the pressed state of GO aerogels in the device", and corresponding characterization should be added to support this point.

After the authors address above issues, the manuscript can be accepted for publication in Nature Communications.

Reviewer #2 (Remarks to the Author):

The revised manuscript resolved all the concerns raised by the reviewers. Thus, this manuscript could be acceptable in the current form.

Reviewer #3 (Remarks to the Author):

In the revised manuscript, the authors have conducted the additional zeta potential and ionic measurements to support the surface charge feature, and also have demonstrated self-healing abilities for the devices. The reusability and long-term performance are valuable for real applications. The quality of the manuscript has been greatly improved and most of the raised issues have been addressed. I recommend its publication in Nature Comm.

We are deeply appreciated to all reviewers' helpful comments. Provided below is our detailed response to each comment/suggestion. The specific changes made to the manuscript to address each point are highlighted in yellow.

Responses to Reviewers' Comments:

Reviewer #1 (Remarks to the Author):

In this revised manuscript, the authors still need to consider the following comments.

1. For the EIS data in Supplementary Fig. 4h, it is too confused to fit the equivalent circuit. At the same time, the test conditions should be stated in the text so that we can analyze the state inside under this condition.

Response: Thanks for the reviewer's precious suggestion.

The equivalent circuit of the osmotic power source with and without RTIL has been fitted in the revised Supplementary Fig. 4h. Electrochemical impedance spectrum (EIS) was carried out at 25 °C under RH of 70%. The Nyquist plot can be fitted with different types of equivalent circuits to obtain the required information. The most commonly used equivalent circuit for porous electrodes¹ is applied as inset schematic in Supplementary Fig. 4h. The $R_s(C_{dl}(R_{ct}(C_p R_p)))W$ equivalent circuit fits very well with the measurement data as shown in Supplementary Table 1, where R_s is the solution resistance (contact resistance), C_{dl} is the double-layer capacitance, R_{ct} is the charge transfer resistance, C_p is the polarization capacitance, R_p is the polarization resistance and W is the Warburg impedance. Addition of RTIL improves not only the solution (contact) resistance but also the charge transfer resistance significantly.

The fitted charge transfer resistances are $5.037 \times 10^7 \Omega$ and $2.091 \times 10^4 \Omega$ for the osmotic power source without RTIL and with RTIL respectively are shown in Supplementary Table 1. The fitted curves (lines) are comparable to the original measurement data in Supplementary Fig. 4h.

The corresponding discussion has been added in the revised manuscript. Detailed

revision is as below.

In the revised manuscript (Page 8, line 155-160):

“Electrochemical impedance spectrum (EIS) is capable of high precision and is frequently used for the evaluation of heterogeneous charge-transfer parameters. The Nyquist plot (Supplementary Fig. 4h) of the osmotic power source with and without RTIL was compared and fitted by the most commonly used equivalent circuit for porous electrodes (inset of Supplementary Fig. 4h). Addition of RTIL improves not only the solution (contact) resistance but also the charge transfer resistance significantly as shown in Supplementary Table 1.”

Supplementary Table 1 Comparison of the EIS fitted results for the osmotic power source without RTIL (Au/AgNO₃/GO/rGO/Au) and with RTIL (Au/AgNO₃/GO/RTIL/rGO/Au).

	Solution resistance (R_s, Ω)	Charge transfer resistance (R_{ct}, Ω)
without RTIL	1.883×10^5	5.037×10^7
with RTIL	2.696×10^4	2.091×10^4

Supplementary Fig. 4h, Nyquist plot of electrochemical impedance spectrum (EIS) of the osmotic power sources with RTIL (Au/AgNO₃/GO/RTIL/rGO/Au) and without RTIL (Au/AgNO₃/GO/rGO/Au) at 25 °C under RH of 70%. The $R_s(C_{dl}(R_{ct}(C_p R_p))W$ equivalent circuit (Inset schematic) was used to fit the power source, where R_s is the solution resistance (contact resistance), C_{dl} is the double-layer capacitance, R_{ct} is the charge transfer resistance, C_p is the polarization capacitance, R_p is the polarization resistance and W is the Warburg impedance.

2. For our previous comment 10, the authors said “the diameter of several nanometers refers to the pressed state of GO aerogels in the device”, and corresponding characterization should be added to support this point.

Response: SEM characterization for the pressed state of GO aerogels in the device was added in the Supplementary Fig. 8.

The corresponding discussion has been added in the revised manuscript. Detailed revision is as below.

In the revised manuscript (Page 13, line 273-275):

“When the GO aerogel was used in the osmotic power device, it was pressed forming nanopores (Supplementary Fig. 8), and the thickness of the aerogels also influences the output performance of the power source.”

Supplementary Fig. 8 SEM image of the pressed GO aerogel (cross section)

After the authors address above issues, the manuscript can be accepted for publication in Nature Communications.

Response: We sincerely thank the reviewer for appreciating and recommending our work for publication in Nature Communications. We have revised the manuscript according to the reviewer’s precious suggestions point by point.

Reviewer #2 (Remarks to the Author):

The revised manuscript resolved all the concerns raised by the reviewers. Thus, this manuscript could be acceptable in the current form.

Response: We are grateful to the reviewer for the thorough review and help to improve the paper.

Reviewer #3 (Remarks to the Author):

In the revised manuscript, the authors have conducted the additional zeta potential and ionic measurements to support the surface charge feature, and also have demonstrated self-healing abilities for the devices. The reusability and long-term performance are valuable for real applications. The quality of the manuscript has been greatly improved and most of the raised issues have been addressed. I recommend its publication in Nature Comm.

Response: We are grateful to the reviewer for the thorough review and help to improve the paper.

References

1. Andrzej Lasia. Electrochemical Impedance Spectroscopy and its Applications Ch. 9, (Springer, 2014).